



# Effects of upwelling duration and phytoplankton growth regime on dissolved oxygen levels in an idealized Iberian Peninsula upwelling system

João H. Bettencourt[1*], Vincent Rossi[2], Lionel Renault[1], Peter Haynes[3], Yves Morel[1], Véronique Garçon[1]

[1]LEGOS, University of Toulouse, CNES, CNRS, IRD, UPS, Toulouse 31400, France
[2]MIO (UM 110, UMR 7294), CNRS, Aix-Marseille Univ., Univ. Toulon, IRD, 13288, Marseille, France
[3]Department of Applied Mathematics and Theoretical Physics, University of Cambridge, England

*Correspondence to:
J. H. Bettencourt (joao.bettencourt@tecnico.ulisboa.pt)
Marine Environment Group
CENTEC - Centre for Marine Technology and Ocean Engineering
Instituto Superior Técnico (Pavilhão Central)
1049-001 Lisboa
Portugal

**Abstract.** We apply a coupled modelling system composed of a state-of-the-art hydrodynamical model and a low complexity biogeochemical model to an idealized Iberian Peninsula upwelling system to identify the main drivers of dissolved oxygen variability and to study its response to changes in the duration of the upwelling season and in phytoplankton growth regime. We find that the export of oxygenated waters by upwelling front turbulence is a major sink for nearshore dissolved oxygen. In our simulations of summer upwelling, when phytoplankton population is generally dominated by diatoms whose growth is largely enhanced by nutrient input, net primary production and air-sea exchange compensate dissolved oxygen depletion by offshore export over the shelf. A shorter upwelling duration causes relaxation of upwelling winds and a decrease in offshore export, resulting in a slight increase of net dissolved oxygen enrichment in the coastal region as compared to longer upwelling durations. When phytoplankton is dominated by groups less sensitive to nutrient inputs, growth rates decrease and the coastal region becomes net heterotrophic. Together with the physical sink, this lowers the net oxygenation rate of coastal waters, that remains positive only because of air-sea exchanges. These findings help disentangling the physical and biogeochemical controls of dissolved oxygen in upwelling systems and, together with projections of increased duration of upwelling seasons and phytoplankton community changes, suggest that the Iberian coastal upwelling region may become more vulnerable to hypoxia and deoxygenation.

## 1 Introduction

Marine hypoxia is an increasing global concern that is exacerbated by climate change, directly through warming-induced increase of stratification and decrease of oxygen solubility, and indirectly by changes in circulation and wind forcing (Levin, 2018). Declining dissolved oxygen (DO) in the world ocean and coastal realm impacts all marine life, from microbes to higher trophic levels (Breitburg et al., 2018), with consequences ranging from ecological adaptations and shifts (Gilly et al., 2013), changes of biogeochemical activity (Wright et al., 2012) to mass mortality events (Diaz and Rosenberg, 2008) and biodiversity restructuring (Vaquer-Sunyer and Duarte, 2008).

Coastal waters are generally eutrophic and characterized by substantial planktonic productivity at the surface which favours oxygen consumption through remineralization of sinking organic matter, leading to low levels of DO in subsurface and near-bottom waters. Highly productive surface waters are typical of coastal upwelling regions and sustain socio-economically important ecosystems. Upwelling systems are thus especially sensitive to deoxygenation (Paulmier and Ruiz-Pino, 2009) and to episodic hypoxia events with deleterious effects on marine life and human activities such as fisheries (Grantham et al., 2004; Hales et al., 2006; McClatchie et al. 2010; Roegner et al., 2011).





The western Iberian Peninsula Upwelling System (IPUS) is the northern branch of the Canary Upwelling System where the intra-annual variability of alongshore winds produce a seasonal upwelling/downwelling cycle (Wooster et al., 1976). In the summer and early fall, the Azores High migrates northward, causing a poleward alongshore wind that forces offshore Ekman transport of surface waters and upwelling of subsurface waters, while downwelling prevails the rest of the year. Long-term studies of the seasonal upwelling in the IPUS have pointed to a weakening of upwelling winds over multidecadal time scales (Sousa et al., 2017), but simulations of future climate indicate an enhancement of the upwelling due to poleward migration of the Azores High and a lengthening of the upwelling season (Miranda et al., 2013; Rykaczewski et al., 2015; Sousa et al., 2017).

During the upwelling season, hydrographic and biogeochemical variability is primarily determined by the wind forcing that controls the inflow of offshore deep water masses onto the shelf (Alvarez-Salgado *et al.*, 1993). DO, in addition of being transported by the Ekman upwelling circulation, is also affected by the turbulent component of the circulation, characterized by sub/mesoscale fronts, filaments and eddies (Bettencourt et al., 2015; Capet et al., 2008; Chaigneau et al., 2009; Marchesiello et al., 2003; Montes et al., 2014; Vergara et al., 2016). These structures have a strong influence on the cross-shore transport and on the vertical redistribution of biogeochemical tracers (Bettencourt et al., 2017; Combes et al., 2013; Gruber et al., 2011; Hernández-Carrasco et al., 2014; Nagai et al., 2015; Renault et al., 2016; Rossi et al., 2013).

From the biological perspective, the planktonic community structure can also affect DO in the water column, as this results from the balance between oxygen production by photo-autotrophs and respiration by both auto and heterotrophs. Changes in the community composition should lead to changes in the DO budget of the continental shelf. In the Iberian upwelling, planktonic blooms tend to be dominated by large cells (micro-phytoplankton) such as diatoms (Moncoiffé et al., 2000; Rossi et al., 2013), that have lower respiration to photosynthesis ratios than dinoflagellates or cyanobacteria (López-Sandoval et al., 2014). Moreover, diatoms growth is thought to be positively correlated with the upwelling-driven nitrate inputs; in contrast, the growth of other planktonic groups can be insensitive to and even limited by newly-upwelled nitrates, especially when colimitation prevails or in the absence of necessary micro-nutrients. Thus, changes in the relative dominance of functional groups in the community can lead to different autotrophy/heterotrophy regimes and total DO levels.

In this paper, we perform a process-oriented study coupling an idealized IPUS configuration of a hydrodynamic model with a low-complexity biogeochemical model centred on oxygen. We focus on the mechanisms of DO change to assess the relative role/importance of physical (upwelling duration) and biogeochemical (functional planktonic community) processes in setting DO levels and distribution. Previous studies have focused on the seasonal and climatological modelling of the planktonic ecosystems and of dissolved oxygen (Marta-Almeida et al., 2012; Reboreda et al., 2014, 2015). Here, the spatio-temporal scales studied are shorter since we are interested in the variations and forcing mechanisms of DO levels in response to successive and short-lived upwelling pulses, commonly observed in the IPUS during the upwelling season. The coupled physical-biogeochemical model is presented in Sect. 2; our results are shown and analysed in Sect. 3. We then discuss our findings and conclude in Sect. 4.

## 2 Materials and methods

### 2.1 The coupled model

The idealized IPUS consist in a meridionally-oriented stretch of continental shelf with constant bathymetry and limited zonally by the coast and the open ocean (Fig. 1). The domain has 300 km of length and width, with a shelf width of 50 km, which is characteristic of the Iberian shelf at 41º N. The resolution of the spatial grid is 500 m, allowing the simulation of summer upwelling seasons up to 90-day duration, with the associated upwelling front dynamics – meso and submesoscale vortices and filaments - within acceptable computing time limits.


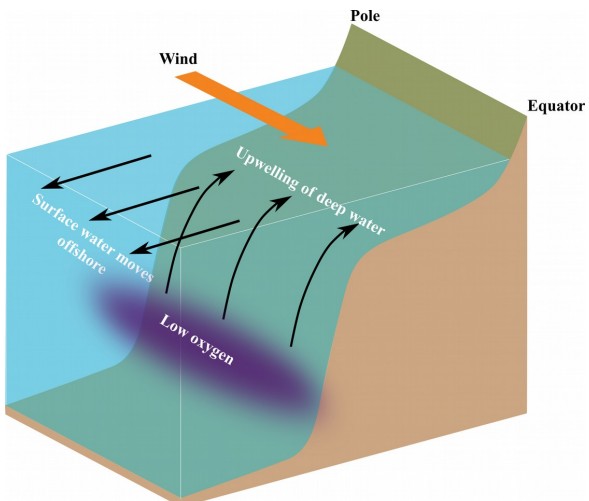

**Figure 1. Idealized coastal upwelling configuration. The domain is periodic in the meridional (pole-equator) direction.**

The physical-biogeochemical model couples the 3-dimensional Regional Ocean Modelling System ROMS (Shchepetkin,

2015; Shchepetkin and McWilliams, 2005), in its Coastal and Regional Ocean Community (CROCO) version (Debreu et al., 2012) to the low-complexity Oxygen-Phytoplankton-Zooplankton biogeochemical model of Petrovskii et al. (2017) and Sekerci and Petrovskii (2015), hereafter SP2015. Oxygen is used as the main model currency since it is central for the functioning of marine ecosystems (Breitburg et al., 1997, 2018). Oxygen is also at the heart of the photosynthesis and carbon fixation metabolic pathways for prokaryotic and eukaryotic algae, being a major electron acceptor (Mackey et al., 2008; Zehr

and Kudela, 2009). and is needed for many biogeochemical reactions, including those involved in remineralization, that constantly occur in the oceanic water column. On top of the oxygen that is produced and consumed within marine ecosystems, a considerable part of dissolved oxygen eventually become gaseous and is then exchanged at the ocean-atmosphere interface, contributing to the estimations that more than one half of atmospheric oxygen is produced in the ocean (Harris, 1986).

ROMS–CROCO is a hydrostatic primitive equation model, formulated in finite differences with time-splitting in the barotropic (fast) mode and the baroclinic (slow) mode. It uses a terrain following vertical discretization in σ coordinates with 40 vertical levels. The prognostic variables are the horizontal barotropic and baroclinic momentum components, temperature and salinity. Free-surface displacement and vertical momentum are diagnostic variables. Horizontal advection of momentum and tracers is calculated from a 3rd order upstream biased scheme. Horizontal mixing is Laplacian, along iso–σ surfaces, with

diffusivity coefficients of 2 $m^2s^{-1}$ and 0.2 $m^2s^{-1}$ for momentum and tracers respectively. Vertical mixing uses the KPP scheme of Large *et al.* (1994). The model is initialized from rest; the initial temperature and salinity distributions are uniform in the horizontal and the vertical profiles are taken from the World Ocean Atlas 2013 climatology (Boyer et al., 2013). Initial fields of the biogeochemical tracers ($O_2$, P and Z) are also horizontally homogeneous and their vertical profiles are obtained from a water column (e.g. 1-dimensional) model version of the $O_2PZ$ model (not shown). The channel is periodic in the meridional

boundaries; the eastern boundary (coast) is modelled as a free-slip wall and the western one (open ocean) is open with radiative boundary conditions for momentum and tracers. Within the westernmost 50 km, a sponge layer is applied with increased horizontal viscosity of 200 $m^2s^{-1}$ and a nudging time-scale of 3 days in order to nudge all prognostic variables to their initial values over the offshore region.  The bottom stress formulation is logarithmic with a roughness height of 0.01 m. The model is forced by a cyclic uniform wind field. To promote the destabilization of the upwelling front, we introduce a

small spatial perturbation to the wind field, whose amplitude is less than 1% of the total wind speed and with characteristic wavelengths of 10 km and 50 km in the alongshore and cross-shore directions, respectively.





In the coupled model, the time evolution of a tracer concentration C is given by:

$$\partial C/\partial t + u\partial C/\partial x + v\partial C/\partial y + w\partial C/\partial z = k\,(\partial^2 C/\partial x^2 + \partial^2 C/\partial y^2 + \partial^2 C/\partial z^2) + F_A + Q_C \qquad (1)$$

where $(u,v,w)$ is the 3D velocity field, $k$ the tracer diffusivity, $F_A$ is the air-sea oxygen exchange flux term (positive when the ocean receives oxygen) and $Q_C$ is the source/sink term for the tracer C, here defined by the biogeochemical model $O_2PZ$ (SP2015). It simulates seven biogeochemical reactions between the three compartments (Fig. 2) as follows:

$$Q_{O2}=AJ(z)f([O2])[P]-u_r([O2],[P])-v_r([O2],[Z])-m[O2] \qquad (2)$$

$$Q_P=l(N)g([O2],[P])-e([P],[Z])-\sigma[P] \qquad (3)$$

$$Q_Z=\kappa([O2])e([P],[Z])-\mu[Z] \qquad (4)$$

In the source/sink term expression for DO (Eq. 2), the term $AJ(z)f([O_2])$ is the rate of $O_2$ production by photosynthesis and is modelled by a Monod parametrization $f([O_2])=c_0/([O_2]+c_0)$, where $c_0$ is a half-saturation constant. The term $J(z) = 1 - exp(\alpha PAR(z)/A)$ represents the light dependency of photosynthesis, with $\alpha = 30$ and the photosynthetically available radiation $PAR(z) = PAR(0)\,exp(-k_w z)$, with $PAR(0) = 355.19$ W/m$^2$ and $k_w = 0.04$ m$^{-1}$.

We distinguish here the terms 'photosynthesis' from 'carbon fixation' (Behrenfeld et al., 2008). Indeed, 'photosynthesis' is the production of ATP and NADPH by the photosynthetic electron transport (PET) chain and these two products are referred to as 'photosynthate'. By contrast, 'carbon fixation' refers to the use of photosynthate by the Calvin cycle to produce simple organic carbon products independently of light. SP2015 chose to highlight these two stages in their model formulation by decoupling carbon fixation from photosynthesis. The respiration by phytoplankton $u_r$ is given by $u_r([O_2],[P])=\delta[O_2][P]/$

$([O_2]+c_2)$, where $\delta$ is the maximum per capita phytoplankton respiration rate and $c_2$ is the half saturation constant. Zooplankton respiration is similar: $v_r([O_2],[Z])=v[O_2][Z]/([O_2]+c_3)$ and the constants $v$ and $c_3$ have the same functions as in $u_r$. A Monod function is a logical formulation to have respiration depending on oxygen concentration since if the fluid environment becomes anoxic, then plankton cannot breathe so that respiration terms $u_r$ and $v_r$ vanish. . The fourth term parametrizes bacterial respiration (oxygen loss due to natural depletion, e.g. remineralization).

In Eq. (3), phytoplankton growth $g([O_2],[P])$ is given by a linear growth term $\alpha([O_2])[P]$ minus intraspecific competition $\gamma[P]^2$, where $\gamma$ is the intraspecific competition intensity. The per capita growth rate is a Monod function $\alpha([O_2])=B[O_2]/([O_2]+c_1)$ where $B$ is the maximum per capita growth rate in the large $[O_2]$ limit and $c_1$ is the half saturation constant. The $l(N)$ term is a Monod function $l(N)=k_1N(\rho)/(k_2 + N(\rho))$, where $N(\rho)$ is a parametrization of nutrient availability based on the nitrate-density ($\rho$) relationships measured during an oceanographic cruise which surveyed the IPUS during upwelling season

(the 2007 MOUTON campaign, see also Rossi et al. 2010; 2013).

SP2015 used a formulation where high oxygen increases phytoplankton growth but limits oxygen production by photosynthesis. The Warburg effect (Turner, 1962) corresponds to the decrease of the photosynthesis rate at high oxygen concentrations so the choice of $f([O_2])$ where high $[O_2]$ limits $[O_2]$ production makes sense. Moreover oxygen stimulates photorespiration which reduces photosynthesis yield. As soon as a water molecule ($H_2O$) undergoes photolysis in the

chloroplast of oxygenic photosynthetic organisms, $O_2$ as well as ATP (Adenosine TriPhosphate) are being produced through both photosystems I and II. The products of photosynthesis (ATP) then enters the Calvin cycle to convert carbon dioxide and other compounds into glucose, that is the food that autotrophs need to grow. As such, we can relate the growth term $\alpha([O_2])$ to $[O_2]$ with a Monod function so as high $[O_2]$ corresponds to high growth rate. The predation term is $e([P],[Z])=\beta[P][Z]/([P]+h)$ where $\beta$ is the maximum predation rate and $h$ is the half saturation constant. Phytoplankton mortality

is $\sigma[P]$. Regarding zooplankton, its feeding efficiency is a function of oxygen concentration $\kappa([O_2])=\eta[O_2]^2/([O_2]^2+c_4^2)$ where $\eta$ is the maximum feeding efficiency and $c_4$ is the half-saturation constant. Zooplankton mortality is given by $\mu[Z]$.





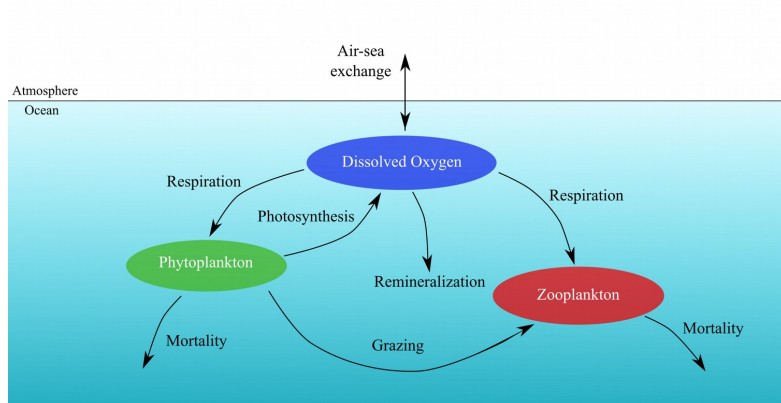

**Figure 2. The Oxygen-Phytoplankton-Zooplankton model of SP2015, adapted to the oceanic environment.**

The parameters of the O₂PZ model were adjusted so that its 0-D version (without advection or diffusion, see table A3) has a

steady state at values representative of the IPUS (see Appendix A).

### 2.2 Simulations

The coupled model is initialized at rest. Biogeochemical tracers are initialized from vertical profiles (Fig. 3(a)) obtained by a water column 1D model of the O₂PZ model that was run for long enough time to reach stable vertical distributions (not shown). Using these profiles instead of climatological initial conditions allows reducing the 3D coupled model drift (not

shown). Initial temperature and salinity profiles (Fig. 3(b)) were taken from the World Ocean Atlas 2013 (WOA13, Boyer et al., 2013) data at 12°W and 41°N.

Our set of simulations (Table 1) is designed to investigate the sensitivity of the coastal DO inventory to upwelling duration and community structure (diatom dominated or not). We choose the duration of upwelling-favourable wind instead of its magnitude as a control factor because it has been shown to be a better predictor of coastal hypoxia (Feng et al., 2012; Forrest

et al., 2011; Zhang et al., 2018). Community structure is set through the growth regime of the primary producers (phytoplankton) with respect to nutrient input. Thus, a diatom dominated regime is simulated using enhanced growth rates due to nutrient input – the *(E)nhanced* runs - while a non-diatom dominated regime is simulated with growth rates *(N)eutral* to and *(L)imited* by nutrient input. We test the sensitivity of our coupled system to the relative dominance of diatoms in the community as they have been suggested to decline over the long-term, being gradually replaced by dinoflagellates and

cyanobacteria (Gregg et al., 2017).

The effects of upwelling season length are studied by running two simulations where the number of upwelling favourable wind cycles are 4 and 9, maintaining a total simulation time of 90 days and considering a diatom dominated phytoplankton community. The wind profiles (Fig. 3(c)) are based on the cyclic wind pulses that are characteristic of the summer / early fall upwelling favourable conditions in the IPUS: 10-day pulses with maximum wind speed of 12 ms$^{-1}$ (Rossi et al., 2013; Torres

et al., 2003a).

**Table 1 Simulations. ECC is the reference simulation. All simulations are run for 90 days.**

| Simulation | P Growth ($k_1$,$k_2$) | Upwelling season |
|---|---|---|
| NCC | Neutral (1,0) | 9 cycles |
| LCC | Limited (1,0.5) | 9 cycles |
| ECC | Enhanced (2,0.5) | 9 cycles |
| ECS | Enhanced (2,0.5) | 4 cycles |


In-situ observations of the IPUS, collected during the MOUTON 2007 campaign (Rossi et al., 2010; 2013), and compiled within the WOA13 (Boyer et al., 2013) are used not only to initialize but also to validate the outputs of our simulations. The

ECC simulation used the wind forcing and phytoplankton growth regimes that most resemble the IPUS region and therefore was chosen as the reference simulation. We compared the full ranges of densities ρ, DO concentrations [O2], and chlorophyll-a concentrations [Chl-a] of the reference simulation to all compiled measurements collected during the MOUTON campaign. We confirm that the coupled model, although of low complexity, reproduces relatively well the ρ-[O2] and ρ-[Chl-a] relationships obtained from in-situ observations, despite a slight underestimation of [O2] and [Chl-a] (Fig. 4).

Note however that the cruise was specially designed to sample high coastal chlorophyll concentrations observed from satellite, so the dataset used in the comparison is biased towards high chlorophyll conditions.

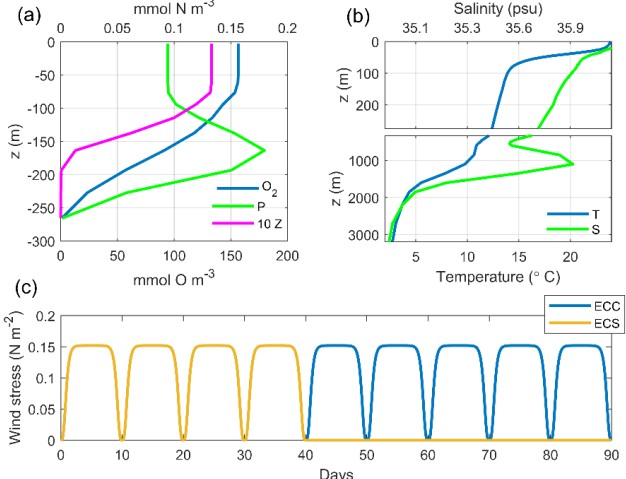

**Figure 3. Initial conditions and wind forcing. (a) Initial profiles of O2, P and Z. (b) Initial profiles of temperature (T) and salinity**
**(S). (c) Wind forcing regimes; ECC: 9 wind cycles; ECS: 4 wind cycles.**

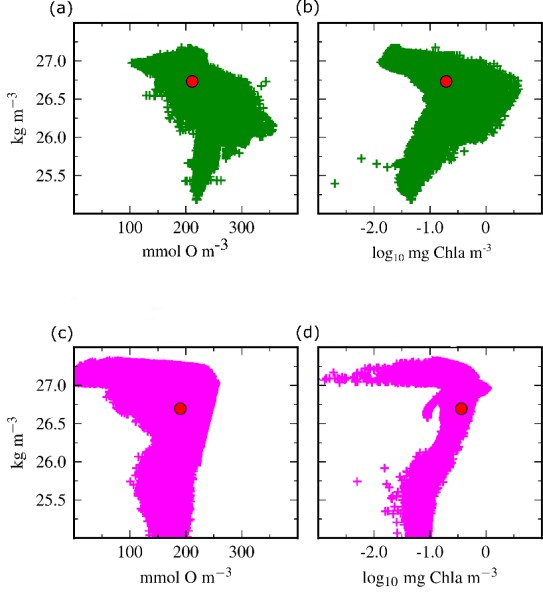

**Figure 4. Validation of ECC simulation. a) Measured density vs. [O2]; b) Measured density vs. [Chla](fluorescence as a proxy for Chl–a); c) Modeled density vs. [O2]; d) Modeled density vs. [Chl-a].**

## 3. Results and discussion

### 3.1 Dissolved Oxygen in the Idealized IPUS

Our coupled model reproduces well the upwelling circulation and the typical biological responses. Indeed, the upwelling of nutrient-rich waters induced by the mean wind-driven circulation promotes the growth of phytoplankton, leading to increased oxygen production by photosynthesis and the subsequent enrichment of oxygen in shelf waters (x < 80 km, Fig. 5(a)), as compared with the lower $[O_2]$ found in offshore waters (x > 80 km). It also shows a low $[O_2]$ cell (~ 50 mmol m$^{-3}$), at the subsurface over the outer-shelf (centred at 40 km and 60 m depth) because of the upwelling of low $O_2$ waters, consistent with the shelf low $[O_2]$ cell (< 200 μmol kg-1) measured during the MOUTON cruise (e.g. Rossi et al. 2013). Similarly, Gutknecht *et al.* (2013) in their modelling study of the Benguela upwelling region found a low $[O_2]$ plume for the climatological month of December at the shelf edge. For the Oregon coast, Hales *et al.* (2006) measured $[O_2]$ of 70-110 mmol m$^{-3}$ in upwelled water at the shelf break, about 200 mmol m$^{-3}$ less than at the surface, which is the same range of the vertical gradient simulated here.

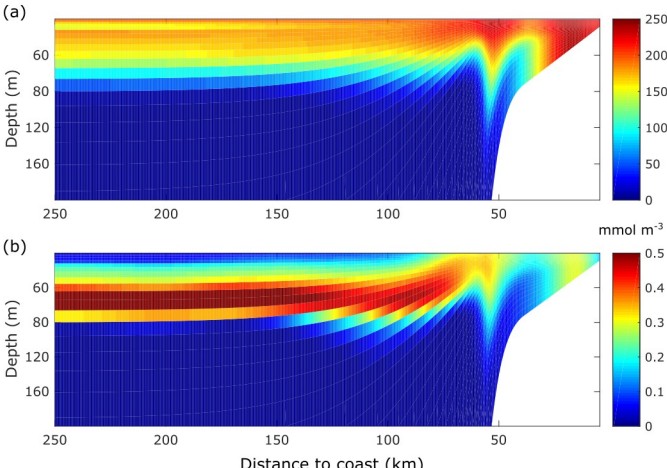

**Figure 5. Mean [O2], [P] fields of the ECC simulation. a) Time mean of along-shore averaged [O2] field. b) Time mean of along-shore averaged [P] field. Time means are computed from day 20 to day 90.**

The dynamics of the ECC simulation (Appendix B), are consistent with those documented by Durski and Allen (2005). A mean Ekman upwelling transport is established, moving subsurface waters upwards near the coast. The eddy induced circulation (Cerovečki et al., 2009; Plumb and Ferrari, 2005) has the opposite sense to the mean circulation and is the strongest in the 50 to 100 km offshore range (Fig. B1). The result of the eddy induced onshore and downward circulation is mainly seen in the subduction of the oxygen rich waters just offshore of the shelf edge, as suggested by the spatial pattern of the eddy-stream-function (Fig. B1).

The spatial coincidence between the high $[O_2]$ and high phytoplankton concentration [P] near the coast (Fig. 5) reflects $O_2$ enrichment caused by photosynthetic production (see Sect. 3.2 for a budget analysis). Since oxygen limits phytoplankton growth in our model, the low $[O_2]$ cell in the coast causes a matching low [P] cell at the same location. Further offshore, subsurface [P] and $[O_2]$ decrease as phytoplankton growth is limited by the lack of nutrients in the euphotic zone.

The subsurface [P] maximum between 60 and 80 meters depth results from the trade-off between sufficient oxygen, light levels and nutrients availability. Consequently, the enhanced phytoplankton growth rate due to nutrients appears to

compensate for the low oxygen levels, thus promoting phytoplankton growth. Moreover, the low [O₂] will maintain the zooplankton stocks at low levels, thus constraining the grazing pressure of zooplankton on phytoplankton.

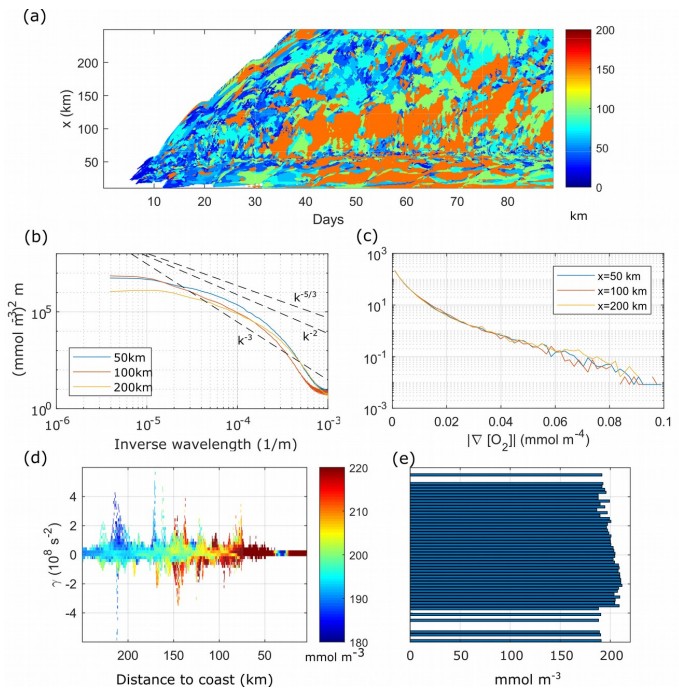

**Figure 6. a) Peak wavelength of surface [O₂] alongshore wavenumber spectrum as a function of time and offshore shore distance. b) Time mean surface [O₂] alongshore wavenumber spectrum at x=50 km, 100 km, and 200 km offshore. [O₂] spectra computed from anomalies with respect to alongshore mean. c) PDF of time mean surface [O₂] gradient norm at x=50 km, 100 km, and 200 km. d) Instantaneous distribution of surface [O₂] by x and γ at day 65. e) Distribution of x-averaged [O₂] by γ at day 65. Time mean taken from day 20 to day 90.**

The dominant length-scales, i.e. the inverse of the wave number where the alongshore [O₂] anomaly field spectrum has a maximum, vary with time and distance to coast (Fig. 6(a)). After day 30, between 50 and 150 km offshore we observe the emergence of dominant length scales of 150 km in an intermittent fashion, in a process that agrees with previous studies of the growth and decay of large mesoscale structures in coastal upwelling systems (Durski and Allen, 2005).

Slopes of time averaged spectra of alongshore [O₂] anomalies at three offshore locations (Fig. 6(b)) show, however, that submesoscale has also a marked effect on the [O₂] field. All spectra are shallower than the k-3 slope of geostrophic turbulence (Charney, 1971), implying that the submesoscale is prominent in our simulations (Capet et al., 2008). The spectrum at x=50 km is the shallowest and the closest to the $k^{-5/3}$ slope in the range $8.10^{-4}<k<10^{-4}$. In the lower k range (k < $10^{-5}$), the spectra at 50 and 100 km have higher energy than the spectrum at 200 km, as the dominant length-scale plot shows the largest dominant length-scales in the 50-150 km cross-shore region.

The Probability Density Function (PDF) of the [O₂] gradient norm (Fig. 6(c)) has a heavy tail distribution, associated with intermittency (Capet et al., 2008; Castaing et al., 1990). The existence of long tails in all PDF's indicates that sharp [O₂] gradients are ubiquitous in the channel.

Turbulence redistributes oxygen through fronts and filaments (strain-dominated regions) as well as vortices (rotation-dominated regions). To analyse this process, we use the Okubo-Weiss criterion γ, which separates turbulent velocity fields in strain-dominated (γ > 0) and rotation-dominated (γ < 0) regions. More specifically, we compute the daily average surface [O₂] for each γ and binned shore distance x.



For day 65, the $[O_2]$ distribution by $(\gamma,x)$ (Fig. 6(d)) reproduces the cross-shore $[O_2]$ gradient of coastal elevated $[O_2]$ and lower offshore $[O_2]$ (the background gradient, see also Fig. 5 (a)), in the range $\gamma\sim0$. Deviations from this background gradient are found along the cross-shore direction, as we move to larger $|\gamma|$, implying that $[O_2]$ anomalies with respect to the mean

surface field are mainly found in filaments and vortices. Since the highest $[O_2]$ levels are found in the shelf region, due to photosynthetic production, their presence further offshore strongly suggests offshore export by turbulent structures. We emphasize that offshore transport of $[O_2]$ rich water results in a net loss of $O_2$ for the continental shelf region.

The $[O_2]$ anomalies found offshore in positive and negative $\gamma$ regions highlight the transport and dispersion of O2 by sub/mesoscale structures from the shelf to the open ocean. We find that although the mean $[O_2]$ as a function of $\gamma$ (averaged

over $x$ in each $\gamma$ bin) remains at $\sim200$ mmol m$^{-3}$ (Fig. 6 (e)), the $[O_2]$ distribution is biased toward negative $\gamma$, as found by Combes *et al.* (2013) for tracers in the California current. Our results are also in line with those of Lovecchio *et al.* (2017) that demonstrate the importance of mesoscale processes for the zonal advection of tracers, with a key role of eddies, especially offshore the upwelling front.

The relative importance of the physical versus the biological processes in controlling coastal oxygen dynamics was analysed

in terms of the volume averaged rate of change of DO due to physics (advection and mixing), biology (photosynthesis, community respiration and remineralization) and air-sea fluxes within a shelf control volume (from the coast to 50 km offshore and from the surface to 140 m depth) and averaged from day 20 to 90. The net enrichment rate, accounting for the physical and biological forcing, is positive (2.26 mmol m$^{-3}$ d$^{-1}$) and so, over time, the coastal oxygen inventory increases. DO concentrations are increased by biogeochemical processes at a rate of 4.27 mmol m$^{-3}$ d$^{-1}$ and by air-sea exchange (3.41 mmol

m$^{-3}$ d$^{-1}$) but are limited by physical processes that remove $O_2$ from the coastal region (-5.42 mmol m$^{-3}$ d$^{-1}$) through lateral advection.

### 3.2 Sensitivity of dissolved oxygen to upwelling duration and phytoplankton growth regimes

The reference simulation (ECC) has shown three major features: the appearance of sub-surface low $[O_2]$ cells because of the upwelling of $O_2$ poor waters, the $O_2$ enrichment of surface levels, and the appearance of an $[O_2]$ gradient between the

biologically-mediated enrichment at the inner shelf and the depleted offshore region. This gradient is, on one hand maintained by the upwelling circulation and, on the other hand, smoothed out by the cross-shore transport of $O_2$ by small-scale vortices and filaments. These effects are driven by the wind regime, that controls the upwelling dynamics and instabilities, and by phytoplankton growth rate, which in turn affects phytoplankton concentrations. We now look at the sensitivity of the coupled model to these two drivers.

The continuous development of filaments and vortices in the ECC run maintains the offshore transport of dissolved oxygen (Fig. 7 (a)). The wind shutdown in the ECS run causes a gradual decrease of turbulence which implies a diminution of the $[O_2]$ redistribution by the cross-shore export physical processes, resulting in the creation of a sharper boundary between the oxygen-rich coastal waters and the depleted open ocean (offshore the upwelling front, Fig. 7 (b)).

The phytoplankton growth regime impacts the mean $[O_2]$ distribution since nutrient neutrality (NCC) or limitation (LCC)

results in the decrease of $[O_2]$, when compared with the ECC run. The largest difference between the NCC and the ECC simulations (Fig. 8(a)), are observed in the coastal region, with maximum deviations from the ECC run up to 100 mmol m$^{-3}$. Offshore of the front, on the other hand, we find an increase of $[O_2]$ with respect to the ECC run, with the difference reaching $\sim50$ mmol m$^{-3}$ at x=250 km. This positive difference is due to the absence of nutrient limitation, so that phytoplankton growth and oxygen production persist in the nutrient depleted surface levels.

The difference between the LCC and ECC cases (Fig. 8(b)) is also the highest in the shelf region, slightly above 100 mmol m$^{-3}$. Offshore of the front (x>50 km), the difference reduces to less than 50 mmol m$^{-3}$, above 80 m depth, vanishing below

this depth. In both cases, there is a signature of the low [O$_2$] upwelling over the slope and shelf edge and the eddy-induced subduction just offshore of the front (x=50 km).

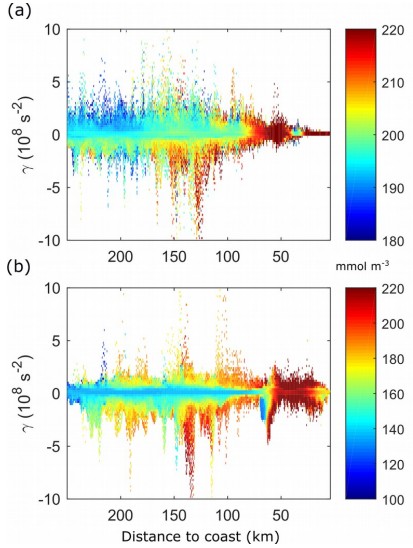

**Figure 7. Time mean distribution of [O$_2$] by x and γ. (a) ECC. (b) ECS. Time mean taken from day 20 to 90.**

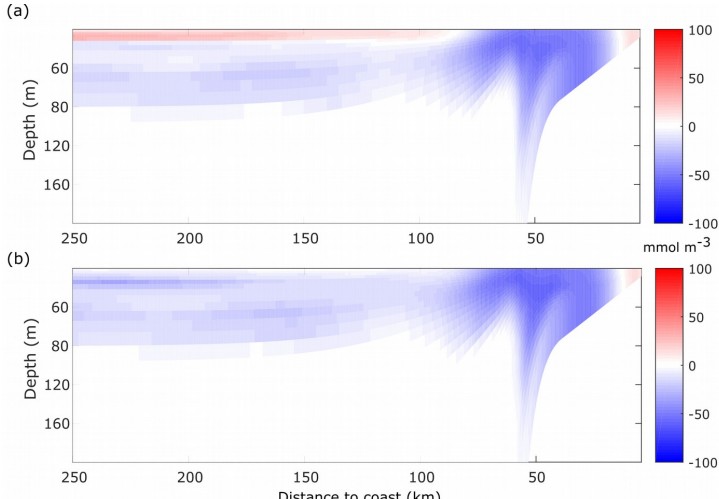

**Figure 8. Difference in time mean dissolved oxygen with respect to the ECC run. (a) NCC. (b) LCC. Time mean taken from day 20 to 90**

When analysing O$_2$ budgets in the coastal control volume, from shore to 50 km offshore and from surface to 200 m of depth, the short upwelling duration simulation (ECS) results in a slight increase of the shelf [O$_2$] temporal mean net rate of change by 6% with respect to the ECC simulation (2.26 to 2.40 mmol m$^{-3}$ d$^{-1}$, Figure 9(a)). Although for ECS every term in the oxygen mean budget is smaller than for ECC, the physical sink decrease is slightly higher than the combined decrease in biological source and sink (Q$_{O_2}$) and air-sea exchange.

The changes in physical sink also occur faster than those in the biological terms. The wind cessation at day 40 causes an immediate decrease in the physical term (Fig. 9(b), lower panel) while only after day 70 can we observe a significant difference between the biological source/sink terms of both ECC and ECS simulations.

Changes in the phytoplankton growth regime modify the physical-biological balance of oxygen in the coastal control volume. The coastal region is net autotrophic ($Q_{O2}>0$) for ECC ($Q_{O2}=4.27$ mmol m$^{-3}$ d$^{-1}$) and ECS ($Q_{O2}=3.99$ mmol m$^{-3}$ d$^{-1}$) but becomes slightly net heterotrophic ($Q_{O2}<0$) when the phytoplankton growth becomes neutral to (NCC, $Q_{O2}=-0.11$ mmol m$^{-3}$ d$^{-1}$) or limited by (LCC, $Q_{O2}=-0.35$ mmol m$^{-3}$ d$^{-1}$) nutrients (Fig. 9(a)). This modifies the [$O_2$] gradients and the [$O_2$] lateral advective fluxes and, as a result, the rate of oxygen loss through lateral advection decreases by about 15% for NCC

and LCC, with respect to ECC. The air-sea exchange of $O_2$ is also affected by the growth regime changes, as lower [$O_2$] will result in increased fluxes.

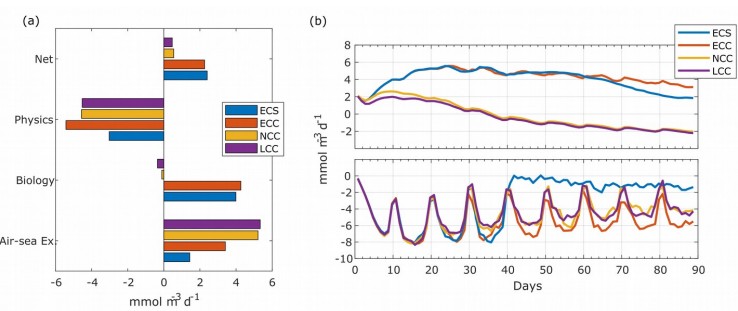

**Figure 9. Dissolved Oxygen budgets in the coastal control volume. (a) Time mean rate of change of dissolved oxygen. Net value is**
**the sum of Physical, Biological and Air-sea exchange source/sink terms. Physical terms are the sum of cross-shore, alongshore and vertical advections as well as horizontal and vertical mixing. Biological term is oxygen photosynthetic production minus community respiration. Time mean taken from day 20 to 90. (b) Time series of average rate of change of dissolved oxygen in the coastal control volume due to biology (top) and physics (bottom).**

Overall, the phytoplankton growth regime has a stronger impact than the wind regime in the DO inventory in the shelf control volume. The NCC and LCC cases result in net rates of change of dissolved oxygen that are 75%-80% lower than in the reference case.

## 4 General discussion and conclusions

We built a coupled physical-biogeochemical model of an idealized seasonal coastal upwelling to study the effect of the
upwelling season length and phytoplankton community structure on dissolved oxygen inventory. While we used data from the Iberian Peninsula upwelling to initialize and validate the range of values simulated by our model, our idealized configuration allows to draw general conclusions about the mechanisms governing the dissolved oxygen levels over the continental shelf. When compared to measurements, our model reproduces well the $O_2$-density relationship as well as upwelling of oxygen-poor waters onto the shelf and the offshore transport of oxygen due to filaments and vortices. While the
addition of air-sea exchange processes as well as our novel nutrient/density parametrization have probably contributed largely to simulate such realistic outputs despite the simple biogeochemical formulation, the coupled system could be further improved. One probably important task, that we keep for future studies, is to include realist subsurface $O_2$ inputs from oceanic currents, as it has been shown to control largely $O_2$ dynamics over the shelf [Montes et al., 2014; Reboreda et al., 2015].





Although there is a clear separation of coastal waters with high [$O_2$] and offshore waters with low [$O_2$], the turbulent lateral transport of high [$O_2$] water disrupts this picture, as oxygen-rich upwelling filaments and vortices generated at the front and then travel offshore. This creates a sink of dissolved oxygen on the shelf that must be compensated by photosynthetic production and air-sea fluxes. In our reference simulation, with repeated upwelling wind cycle, the net [$O_2$] rate of change was positive, owing to the predominance of the photosynthetic oxygen production and air-sea fluxes over the lateral

transport of oxygen, that acts to remove it from the shelf region. When considering a shorter upwelling season (the ECS case with four 10-day upwelling wind cycles followed by wind cessation) reinforces somewhat the oxygen enrichment trend. When phytoplankton community is dominated by groups that show nutrient limitation or neutral growth (LCC and NCC cases), the biological source weakens and the shelf becomes net heterotrophic. The physical sink is also affected, producing a weaker loss of dissolved oxygen in the coastal upwelling. Air-sea exchange then maintains the (much lower) net oxygen

enrichment trend

Our findings that the oxygen inventory net rate of change is inversely proportional to the duration of upwelling season is consistent with other studies (Adams et al., 2013; Siedlecki et al., 2015; Zhang et al., 2018) and, as the upwelling season duration in the IPUS is thought to increase in the future (Miranda et al., 2013; Wang et al., 2015), it suggests increased vulnerability of coastal regions to hypoxia and long-term deoxygenation. The result of lower phytoplankton growth rates

driving the shelf region from autotrophy to heterotrophy also points to an increased risk of loss of dissolved oxygen if trends for community shifts towards smaller or less efficient phytoplankton (Gomes et al., 2014; Gregg et al., 2017; Head and Pepin, 2010; Zhai et al., 2013) are specifically confirmed for the IPUS.

**Code/Data availability**

Simulated data and computer codes are available upon request.

**Appendix A. Determination of O₂PZ model parameters.**

The biogeochemical component of the 3D coupled model (Eqs. (1), (2)-(4)) is a system of ODE's that computes the local time evolution of [$O_2$], [P] and [Z]. The ODE system accounts for the processes that move $O_2$, P and Z between different reservoirs. Each ODE of the model then expresses the input and output of the species from the specific reservoir. The dynamical properties the ODE's will therefore influence the global behaviour of the coupled physical-biochemical

model. When using coupled models such as (1), it is good practice to work in the vicinity of a known co-existence stable fixed point, i.e a solution $X^+ =([O_2]^+,[P]^+,[Z]^+) \neq (0,0,0)$ to Eqs. (2)-(4) that doesn't change with time and attracts nearby system trajectories. This guarantees that, when removing the advection and diffusion terms, the system will evolve to a constant state whose properties are already established (e.g. positive concentrations). The fixed points of the $O_2PZ$ system are the states $X^+$ where

*$d[O_2]^+/dt=d[P]^+/dt=d[Z]^+/dt=0$*                                           (A1)

Since the system is nonlinear, solution of Eq. (A1) is not straightforward and approximate methods are then a good choice. Since the model will be used to study $O_2$ dynamics in coastal upwelling systems in the Iberian Peninsula western shelf/slope, it is logical that the model uses initial conditions that are characteristic of this region. The characteristic state $X^w = ([O_2]^w,$ $[P]^w,[Z]^w)$ may be taken from climatological dataset, in-situ observations, remotely-sensed data, etc. as long as it is

representative of the actual values of ([$O_2$],[P],[Z]) in that region. Regarding the position of $X^w$ in the state space, a simple proposition is that $X^w$ is close to a coexistence steady state $X^0$. This means that, if the model is initialized at $X^w$ it will converge to $X^0$ and will not evolve to an extinction state or other steady states with unrealistic or uncharacteristic values of





($[O_2]$,[P],[Z]) for the region. This is the approach adopted here to study the fixed points and determine the values of the parameters of the system and we adopt the characteristic values shown in Table A1 for the Iberian Peninsula upwelling.


**Table A1. Characteristics values of $O_2$, P and Z for the western Iberian Peninsula.**

| Variable | Value |
|---|---|
| $[O_2]^w$ | 0.255 mmol O L$^{-1}$ |
| $[P]^w$ | 0.063 mmol O m$^{-3}$ |
| $[Z]^w$ | 0.017 mmol O m$^{-3}$ |

The concentration of DO is taken from the climatological mean at 12°W, 41° N and 10 m depth. The phytoplankton concentration is obtained from ocean color images of chlorophyll a in the west Iberian shelf at approximately the same

position of the DO reading, after converting it to nitrogen concentration using a ratio of 1.59 between Chl a and nitrogen content (Montes et al., 2014). Finally, the zooplankton concentration is found by using a phytoplankton to zooplankton ratio of 3.6, taken from SP2015 for the case of a stable co-existence state of the nondimensional model. For some parameters we adopt the values that are used elsewhere in the literature. To find the most adequate values of the remaining parameters, we search for the parameters that make possible the propositions above regarding the stable nature of $X^w$. To proceed, we

consider first the system obtained by setting [Z] = 0.

Applying the fixed point condition (A1), we obtain two equations relating [P] and $[O_2]$:

$$P_1([O_2]) = (m[O_2]([O_2]+c_0)([O_2]+c_2))/(Ac_0c_2+(A-\delta)c_0[O_2]-\delta[O_2]^2) \tag{A2}$$

$$P_2([O_2]) = 1/\gamma(Bc/([O_2]+c_1) - \sigma) \tag{A3}$$

The fixed points of the system are given by $P_1 = P_2$ in the positive quadrant of the ($[O_2]$, [P]; [Z] = 0) plane, in addition to the

extinction steady state. Explicit computation of the fixed points is not possible for the $[O_2]$-[P]-[Z] system, but we can derive relationships of the form [Z] = f ($[O_2]$,[P]) that satisfy (A1) from the oxygen and phytoplankton equations, as follows:

$$Z_1([O_2],[P]) = ([O_2] + c_3)/\nu[O_2] ((Ac_0[P])/([O_2]+c_0)-(\delta[O_2][P])/([O_2]+c_2)-m[O_2]) \tag{A4}$$

$$Z_2([O_2],[P]) = ([P]+h)/\beta[P] (B[O_2][P]/([O_2]+c_1)-\gamma[P]^2-\sigma[P]) \tag{A5}$$

The zooplankton equation does not produce a similar relationship. Instead, analytical manipulation of it let us derive the

following expression:

$$[Z]=0 \lor [P]=P_2=hM([O_2])/(1-M([O_2])), \tag{A6}$$

where $M([O_2])=\mu/\beta \, ([O_2]^2+c_4^2)/\eta[O_2]^2$. The fixed points of the $[O_2]$-[P]-[Z] system are the locii of intersection of Eqs. (A4)-(A6) in the ($[O_2] > 0$;[P] > 0;[Z] > 0) octant. Note that the terms in parenthesis in Eqs. (A4), (A5) are the equations of the oxygen-phytoplankton system. In fact, if [Z] = 0, Eqs. (A4), (A5) reduce to Eqs. (A2), (A3), as expected.

There are 11 parameters that relate $O_2$ with phytoplankton and zooplankton. It is a considerable amount of free parameters to handle. To constrain the choice of values for this set of parameters, making it easier to tune the model, we introduce the following three assumptions, regarding the model behaviour at $X^w$:

1. At $X^w$, zooplankton growth should not be hampered by the value of $[O2]^w$. This requires that the feeding efficiency term is at or near its maximum h.

2. At $X^w$, the phytoplankton growth rate should be near or at its maximum value of B.

3. At $X^w$, phytoplankton $O_2$ production rate is significantly larger than $O_2$ respiration rate.

These assumptions are adopted to facilitate a co-existence state at $X^w$, by not limiting the ability of zooplankton to feed on phytoplankton and at the same time by facilitating phytoplankton growth to feed zooplankton and to produce $O_2$ in a greater amount than it respires.

Since the position of the fixed points of the system is given by the intersections of the isoclines $Z_1$, $Z_2$ and $P_Z$, they should intersect at or near ($[O_2]^w$; $[P]^w$) and the fixed point at the intersection should be stable. This can be achieved by changing the parameters of the system in order to move the isoclines in phase-space, and at the same time check the sign of the real part of


the eigenvalues of the system Jacobian matrix at the intersection to identify the stability type of the fixed point. In Fig. A1, we plot the three isoclines in the plane v = 0 for different values of the parameters A, B and h. It shows how the isoclines

change with the changing parameters. Since both $Z_1$ and $Z_2$ depend on [Z], we must look at the 3D phase space to see where the isoclines intersect (Fig. A2).

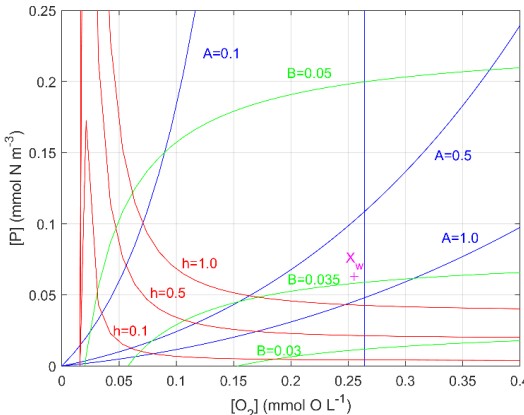

**Figure A1. Isoclines of the [O₂]-[P] system for selected values of A, B and h. Blue: P₁; Green: P₂; Red: Pz.**

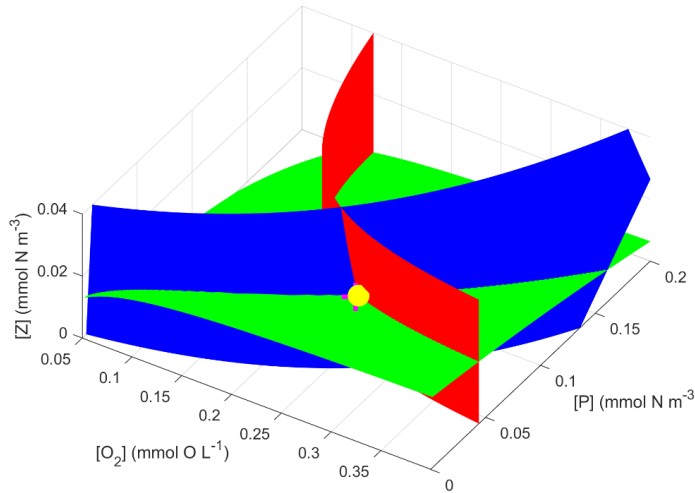


**Figure A2. Isoclines of the [O₂]-[P]-[Z] system. Blue: $Z_1$; Green: $Z_2$; Red: $P_Z$. Yellow: $X^0$. To build this figure , we set up a grid of ([O₂]ᵢ; [P]ⱼ); i; j = 1;…;N values and plotted $Z_1(i; j)$ and $Z_2(i; j)$ from Eqs. (A4) and (A5). The $P_Z$ isocline is different since it does not depend on [Z], so we computed $P_Z(i) = hM(c_i) / (1-M(c_i))$ with M defined in the text, and extended it vertically up.**

To find the intersection point, that corresponds to the co-existence steady state $X^0$, we computed the distance $d_{12}(i; j) = Z_2(i;$

j)- $Z_1(i; j)$, so that the intersection $Z_2 = Z_1$ is given by $d_{12} = 0$. The zeros of $d_{12}$ were identified approximately by choosing, for each [O₂]ᵢ, the value of [P]ⱼ that minimized $d_{12}(i;j)$. To find $X^0$, we used a similar procedure to find the point where this line intersects $P_Z$. This procedure is approximate, and its accuracy varies with N. By performing tests for several N, we found that N = 300 gave enough accuracy. By varying A, B and h, the intersection ($X^0$) can be made to approach $X^w$. Although other parameters may be varied instead of these three, we found that these are the ones whose variations are easier to judge in

terms of displacement of the isoclines. We note that, of the three, only h is one of the "fixed" parameters that was found in





the literature, but since it is quite easy to move $P_Z$ by changing its value, we decided to use it as a tuning parameter. The result of this exercise is that the co-existence fixed point $X^0$ is located near $X^w$, with coordinates shown in Table 2 and is a stable spiral point with eigenvalues of the Jacobian $\lambda_1 = -0.0847$ and $\lambda_{2,3} = -0.0009 \pm 0.0221i$.

**Table A2. Coordinates of the co-existence fixed point $X^0$.**

| Variable | Value |
|---|---|
| $[O2]^0$ | 0.261 mmol O $L^{-1}$ |
| $[P]^0$ | 0.052 mmol N $m^{-3}$ |
| $[Z]^0$ | 0.018 mmol N $m^{-3}$ |

**Table A3. Parameters of the 0-D SP2015 biogeochemical model.**

| Parameter | Value | Units | Description |
|---|---|---|---|
| A | 0.54 | $d^{-1}$ | Environmental effects in rate of $O_2$ production inside phytoplankton cells. |
| $c_0$ | 0.255 | mmol O $L^{-1}$ | Half-saturation constant for $O_2$ production. |
| $\delta$ | 0.2 | $d^{-1}$ | Maximum per capita phytoplankton respiration rate. |
| $c_2$ | 0.255 | mmol O $L^{-1}$ | Half-saturation constant for $O_2$ respiration by phytoplankton |
| $\nu$ | 0.35 | $d^{-1}$ | Maximum per capita zooplankton respiration rate. |
| $c_3$ | 0.255 | mmol O $L^{-1}$ | Half-saturation constant for $O_2$ respiration by zooplankton |
| m | 0.03 | $d^{-1}$ | Rate of oxygen loss due to natural depletion. |
| B | 0.055 | $d^{-1}$ | Maximum per capita phytoplankton growth rate. |
| $c_1$ | 0.017 | mmol O $L^{-1}$ | Half-saturation constant for phytoplankton growth. |
| $\gamma$ | 0.1 | mmol O $L^{-1}$ $d^{-1}$ | Intensity of intra-specific competition. |
| $\beta$ | 0.9 | $d^{-1}$ | Maximum zooplankton predation rate. |
| H | 0.8 | mmol N $m^{-3}$ | Half-saturation constant for phytoplankton predation. |
| $\sigma$ | 0.027 | $d^{-1}$ | Phytoplankton mortality rate. |
| $\eta$ | 0.75 | - | Maximum zooplankton feeding efficiency. |
| $c_4$ | 0.255 | mmol O $L^{-1}$ | Half-saturation constant for zooplankton feeding efficiency. |
| $\mu$ | 0.025 | $d^{-1}$ | Zooplankton mortality rate. |

**Appendix B. Dynamics of the ECC simulation**

We analyze the dynamics of the ECC simulation in terms of turbulent energy transfers, mean and eddy–induced circulations
and dominant turbulent length scales. Mean quantities refer to the along-shore average, e.g., for the cross-shore velocity $u$:

$$<u(x,z,t)>=L^{-1} \int u(x,y,z,t) \, dy \qquad \text{(B1)}$$

where $L$ is the along-shore length of the channel. Perturbations are the departures from this mean: $u' = u - <u>$ and $<u'>=0$. The mean kinetic energy transfer term is $cke =-\rho_0(<v_x><u'v'>+<v_z><w'v'>+<u_x><u'u'>+<u_z><w'u'>)$, where $\rho_0$ is the reference density, $v$ and $w$ the along-shore and vertical velocities and a subscript denotes partial differentiation. The
perturbation potential energy transfer term is $cpe = -g<w'\rho'>$.

In the ECC case $cpe$ remains positive, increasing after the wind intensification part of the wind cycle and decreasing when the wind relaxes (Fig. B1a), while $cke$ oscillates between positive and negative values. The mechanism for cke growth seems to be the coalescence of disturbances into a single large eddy that is subsequently perturbed by rapidly emerging small scale patterns (Durski and Allen, 2005). Anti-correlation between $cpe$ and $cke$ is observable at days 35, 47, 53 and 60 (Fig. B1a),
suggesting that $cke$ evolution is related to nonlinear wave–wave interaction rather than to wave–mean flow interaction (Durski and Allen, 2005). For sustained winds, like the ECC wind cycle with short wind intensification and relaxation periods, the effects of this interaction are significant since the start of the simulations (Durski et al., 2007).
Initially, the surface density peak length-scale $l_e$, i.e. the most energetic scale, is homogeneous, and $l_e$ is equal to the domain length (Figure B1b).


Shortly after the start, small-scale (O(10km)) fluctuations appear close to shore, then grow in length and spread offshore. From day 20 onwards, $l_e$ fully develops and ranges from 20 to 160 km across shore, although until day 36 there are still regions where density is rather homogeneous.

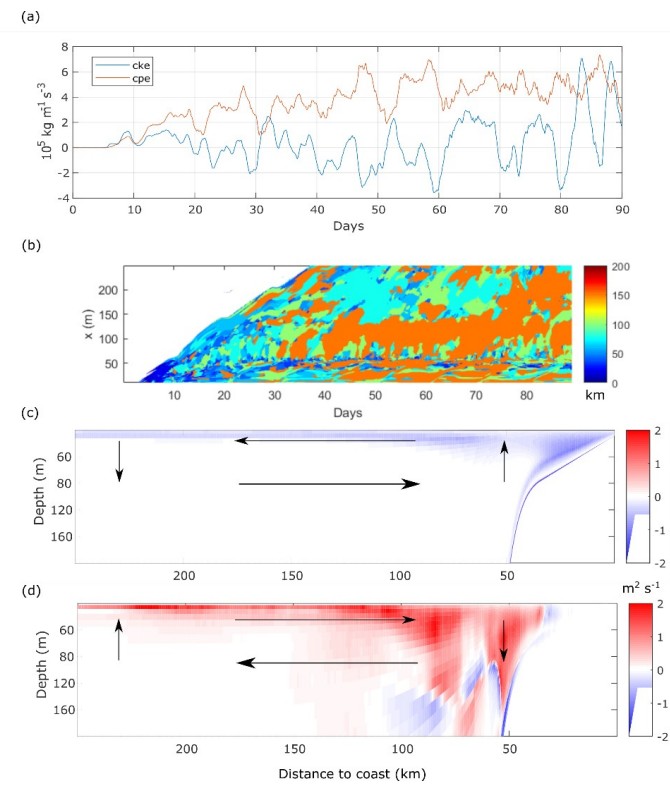

**Figure B1. a) Time series of volume averaged perturbation energy source terms; cke source term (green) and cpe source term (blue); volume average in the whole domain. b) Peak wavelength of surface density anomaly alongshore wavenumber spectrum as a function of time and offshore distance. c) Mean stream-function over days 20 to 90. d) Eddy stream-function over days 20 to 90. Mean stratification is shown as black dash-dot contours in panels c and d. Arrows in panels c and d represent a schematic view of the circulation.**


Offshore the upwelling front (roughly from 50 km) we observe the emergence of two distinct regions in the across-shore direction: a region adjacent to the front where large scale structures are often dominant ($l_e \sim 150 - 160$ km), and further offshore where smaller-scale structures are dominant ($l_e < 100$ km). The cross-shore extent of the first region appears to be related to the excursions in the *cke* time serie (Fig. 3a), judging by the coincidence of the largest fluctuations of *cke* with the

widest swaths of $l_e \sim 150 - 160$ km (days 45, 60, 68 and 80). The second region further offshore seldom exhibits $l_e > 100$ km as it is usually populated by filaments emerging from the front with contrasted density and by vortices resulting from the roll up of filament tips.

The wind-induced mean circulation in the *x-z* plane is $\Psi = -\tau/\rho_0 f$ where $\tau$ is the applied wind stress and $f$ the Coriolis parameter. The eddy stream-function is $\psi = (<v'b'><b'_z> - \lambda^2 <w'b'><b'_y>)/(\lambda^2 <w'b'><b'_y> + <b'_z>)$, where $b$ is buoyancy

and $\lambda = 1000$ is a stretching constant that accounts for the large aspect ratio of oceanic flows (Nagai et al., 2015; Plumb and Ferrari, 2005). The wind-induced mean circulation in the vertical cross-shore plane (Fig. B1c) is stronger at the surface and near the coast and weaker in the interior. A typical upwelling circulation is simulated: Ekman transport moves surface waters offshore and upwards near the coast giving rise to the mean upward tilting of the isopycnals. The eddy induced stream-





function (Fig. B1d) represents the component of the flow across the mean isopycnals (Cerovečki et al., 2009; Plumb and Ferrari, 2005), opposes the mean circulation and can surpass it. In agreement with Cerovečki et al. (2009) we find that the eddy stream-function is stronger at the diabatic surface layer, in a thin layer (up to 20 m depth) with strong shoreward circulation 100 km offshore, deepening to 150 m at 75 km offshore. There is another strong eddy downwelling cell, on the shelf–slope transition, that goes down to 160 m depth.

The submesoscale and mesoscale turbulence is sustained by energy transfer from the potential energy field to the turbulent
kinetic energy field, through baroclinic conversion (Durski and Allen, 2005; Marchesiello et al., 2003). In agreement with previous modelling studies (Capet et al., 2008; Durski and Allen, 2005), the global picture is thus that the intermittent wind bursts act as a recharge for the mean potential energy which is then released to eddy kinetic energy through baroclinic instability. When the wind decreases, the eddy flux induces a rapid re-stratification of the upper layer (Capet et al., 2008).

**Author contributions**

JHB, VR and VG design the study with input from LR, YM and PH. JHB performed the simulations. JHB, VR and VG analysed the results and prepared the draft manuscript. All authors critically reviewed, commented and prepared the final manuscript.

**Acknowledgements**

This work is part of the TEASAO project, funded by "IDEX attractivity chairs" program from University of Toulouse. JHB
acknowledges post-doctoral fellowship financial support from the TEASAO project and CENTEC (University of Lisbon) for hosting him during his stays. VR acknowledges fruitful discussions with M. Marta-Almeida and travel support from INSU-CNRS. The MOUTON 2007 campaign was carried out as part of a SHOM (Service Hydrographique et Océanographique de la Marine) project and we acknowledge the competence and investment of the SHOM technical staff who took part in it.

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
