# Peer review of "Effects of upwelling duration and phytoplankton growth regime on dissolved oxygen levels in an idealized Iberian Peninsula upwelling system"

_Nonlinear Processes in Geophysics, 2019_

## Referee Comment (RC1) · Anonymous Referee #1 · 21 Nov 2019

This article analyses some important physical and biological factors of dissolved oxygen variability in an oceanic upwelling coastal zone. For this they couple a hydrodynamical model of the Iberian Peninsula with a "simple" biogeochemical model considering phytoplankton, zooplankton and dissolved oxygen interactions, and air-sea oxygen exchange flux. A number of results is discussed, in particular, that oxygen net rate of change is inversely proportional to the duration of the upwelling season.

This is an excellent work, very well written (with almost no typos) that helps understanding timely questions concerning the biological and physical factors most influencing dissolved oxygen in the oceans. The numerical work, despite its complexity, is very detailed and properly structured, analysing and discussing in detail the different biological and physical factors. I strongly recommend this article for publication after the authors have addressed some minor changes that I hope will help the reader to follow better this article: i) There is no mention to the numerical method to integrate the advection-reaction-diffusion Eq(1). Is it Eulerian, semi-Lagrangian, other? What about numerical diffusion, is it relevant? ii) Contrasting with this, Appendix A was very detailed. Is it necessary to include so many details to obtain a numerical fixed point when standard softwares (mathematica, matlab,...) makes the work easily? In any case I found this Appendix difficult to follow and I would ask the authors to smooth it out. iii) I think it can be helpful for the discussion if you show the full set of advection-reaction-diffusion equations with all the terms and not in different pieces (Eq (3-4), and all the discussion below). In particular, the way the authors address the air-sea exchange flux term was unclear for me. iv) After reading several times, I do not understand Fig. 6e. Please clarify its meaning. Is it really helpful? v) In the discussion/conclusions section I missed some discussion on the relevance of the results of this work on permanent marine oxygen minimum zones.

---

## Author Comment (AC1) · 21 Feb 2020

We thank the Referee for his/hers comments that helped to greatly improve our manuscript. Below, we provide a point-by-point reply to the referee's comments.

Ref. #1: There is no mention to the numerical method to integrate the advection-reaction-diffusion Eq(1). Is it Eulerian, semi-Lagrangian, other? What about numerical diffusion, is it relevant?

Auth. : Eq (1) is solved online by the ROMS model itself, i. e., concurrently with the hydrodynamic equations, following an Eulerian approach. The model splits the advective part in horizontal advection that is solved by a 3rd order upstream method and vertical advection that uses a 4th order centered method. The horizontal diffusive operator is harmonic and the reaction term is solved by an iterative method, with three fractional time steps to achieve implicit discretization of the biological interactions. The 1D model is solved in the same manner as we used the 1D version of ROMS to perform the simulations. Since the numerical schemes chosen to solve the equations are known to be over diffusive, we did not include explicit diffusion in the model; hence, numerical diffusion is the only source of diffusion in the simulations.

Ref. #1: Contrasting with this, Appendix A was very detailed. Is it necessary to include so many details to obtain a numerical fixed point when standard softwares (mathematica, matlab,...) makes the work easily? In any case I found this Appendix difficult to follow and I would ask the authors to smooth it out.

Auth. : We have dropped the details of the fixed point search and we have considerably shortened Appendix A to make it easier to follow.

Ref. #1: I think it can be helpful for the discussion if you show the full set of advection-reaction- diffusion equations with all the terms and not in different pieces (Eq (3-4), and all the discussion below). In particular, the way the authors address the air-sea exchange flux term was unclear for me.

Auth. : We have improved the way we present the model equations in the new version of the manuscript. All terms pertaining to dissolved O2, P and Z are now shown as one fully-developed differential equation for each species : please see equations (1) – (3) in page 4 of the MS. We have also rewritten the text and developed our explanations to better describe the Air-Sea exchange term.

Ref. #1: After reading several times, I do not understand Fig. 6e. Please clarify its meaning. Is it really helpful?

Auth. : Figure 6e displays the alongshore averaged concentrations of O2 for each binned value of the Okubo-Weiss criterion (OKW) . It is computed from the data of figure 6d by averaging the values in the map "horizontally", that is along the x-axis for a constant OKW level. It emphasizes the result that, on average, there are higher O2 concentrations in the rotation-dominated regions (OKW<0) as compared against strain-dominated regions (OKW>0); this information is not readily observable in figure 6d.

Ref. #1: In the discussion/conclusions section I missed some discussion on the relevance of the results of this work on permanent marine oxygen minimum zones

Auth. : Thank you for this pertinent suggestion; we have added a few words about this topic in the discussion section. The revised discussion now reads as: "In more permanent and extended OMZs, changes in physical forcing (e.g. wind regime) and/or biological structure can have dire implications for marine ecosystems and fisheries management. Indeed, the measured expansion of the Tropical Atlantic and Pacific OMZs (Stramma et al., 2008) implies a shallower upper boundary of the low O2 core and an increased sensitivity of the shelf oxygen levels to upwelling winds, which are indeed predicted to intensify with climate change (Sydeman et al., 2014). Future work could be focused on understanding this high sensitivity and on anticipating how it would affect the stability of the coupled-system in a changing climate."

Sydeman, W.J., García-Reyes, M., Schoeman, D.S., Rykaczewski, R.R., Thompson, S.A., Black, B.A., Bograd, S.J., 2014. Climate change and wind intensification in coastal upwelling ecosystems. Science 345, 77–80. https://doi.org/10.1126/science.1251635

---

## Referee Comment (RC2) · Anonymous Referee #2 · 12 Mar 2020

In the manuscript, a configuration of the CROCO model is discussed, consisting of a simplified version of the study region, using a periodic channel, with periodic winds favourable to the coastal upwelling, disturbed to originate mesoscale structures (eddies and filaments). In particular, emphasis is placed on studying the response of the system to explain the physical mechanisms of the variability (and offshore export) of dissolved oxygen and the phytoplankton growth. The role of oxygenated water export associated to frontal turbulence is identified as the main dissolved oxygen sink near the coast. It is argued that primary production and exchange with the atmosphere represent mechanisms that compensate the dissolved oxygen sink for the open ocean. While I generally agree with the comments of Referee #1, in terms of the quality of the manuscript, and its relevance of publication, there are a number of points that I will refer to, which should be addressed before the final publication. On the one hand, the configuration used is highly idealized, and the practical application to the Iberian system is not straightforward. Provided the simplifications that were done related the idealized configuration, and the theoretical approach, (quite difficult to read to oceanographers not specialized in theoretical oceanography) I have some doubts about the practical applicability of the results, and the interest that it may have for the oceanographic community of the region, devoted to the study of biogeochemical processes, (constituted mainly by chemical oceanographers).

Summarizing, the points below should be addressed:

1 - Introduction: Paragraph 1. The Iberian region is not an example of a region of marine hypoxia, and that should be referred to in the manuscript. References to deoxygenation (line 40) do not make much sense in the scope of this manuscript. 2 - The authors barely refer to the observational literature related to the oxygen cycle in the Iberian region, limiting themselves to cite Rossi's articles (2010, 2013), which consist of a single campaign, not particularly devoted to that subject. There is some observational literature including the oxygen cycle that I think should be cited, as it is relevant. (Hint: Alvarez-Salgado, X.A, Perez, F.F, Castro, C.G. , all of them in IIM-CSIC-Vigo, Spain) On the other hand there are oceanographic processes not solved by the model that influences the oxygen cycle, related to processes in the benthic layers, and on remineralization processes that influence the oxygen cycle, and that are not referred to. 3 - Lines 62-64 need a reference that support those statement. 4 - Is the implementation of the Biogeochemical (BGQ) module done in CROCO model, or was implemented from scratch?. This is not clear in the ms. In that case, justify why none of the other BGQ modules was not used. 4- Is not clear the interaction with the atmosphere in the BGQ module, concerning O2 atmosphere - ocean exchange. 5- Laplacian mixing

along sigma surfaces tends to generate spurious currents. Why you did not use (rotated) mixing along Geopotential? 6 - Is the wind only alongshore?, please specify.(l 109). 7 - It should be clearer to the reader why nutrients are not introduced specifically, and what is the advantage of doing it in an O2PZ model instead of O2NPZ? 8 - The 12m/s wind pulses although exist in particular years (like the MOUTON campaign) are too high and unrealistic for the Western Iberia region. Justify better why such high values. Some sensitivity studies was done to this value? 9- I don't understand why in the initial vertical profile, (Fig 3a) the 02 drops to zero when in reality the O2 at 300m is closer to 230 mmol O / m3. The range of values in the region above central waters is 220-290 mmol O / m3. 10 - Claim that figure 4 represents a validation seems too optimistic to me. I can't understand what the red dot represents in the figures. 11 - Using logarithms in Chla completely distorts the comparison, because the observations are showing values of 0-10 mg Chla.m-3 between isopycn 26-27 as the model rarely goes beyond 1 (at isopycn 27). Similar comment arises for O2. If you claim those results are validated, then, any result is validated. If you insist in keeping this part as it is, I suggest that you remove the word validation, perhaps use 'qualitative comparison'. 12-Figure 5, I can understand the effect of upwelling at x= 40km that is related to shelf upwelling, but how do you interpret the dome at 60-70km (approximately)? To use the references (l 200-204) to validate the averages of O2 seems to me to be an over-optimistic approach. Again you can say that it is a simplified and idealized model, but do not try to convince the reader that it is validated, because readers with observational background would hardly understand it. 13- Lines 210 - 215 seem more like a discussion, and their presence is not understood there.

---

## Author Response (AR1)

**Effects of upwelling duration and phytoplankton growth regime on dissolved oxygen levels in an idealized Iberian Peninsula upwelling system**

João H. Bettencourt , Vincent Rossi , Lionel Renault , Peter Haynes , Yves Morel , Véronique Garçon

**Author's reply to Referee #1**

We thank the Referee for his/hers comments that helped to greatly improve our manuscript. Below, we provide a point-by-point reply to the referee's comments.

Ref. #1: *There is no mention to the numerical method to integrate the advection-reaction-diffusion Eq(1). Is it Eulerian, semi-Lagrangian, other? What about numerical diffusion, is it relevant?*

Authors : *Eq (1) is solved online by the ROMS model itself, i. e., concurrently with the hydrodynamic equations, following an Eulerian approach. The model splits the advective part in horizontal advection that is solved by a $3^{rd}$ order upstream method and vertical advection that uses a $4^{th}$ order centered method. The horizontal diffusive operator is harmonic and the reaction term is solved by an iterative method, with three fractional time steps to achieve implicit discretization of the biological interactions. The 1D model is solved in the same manner as we used the 1D version of ROMS to perform the simulations. Since the numerical schemes chosen to solve the equations are known to be over diffusive, we did not include explicit diffusion in the model; hence, numerical diffusion is the only source of diffusion in the simulations.*

Ref. #1: *Contrasting with this, Appendix A was very detailed. Is it necessary to include so many details to obtain a numerical fixed point when standard softwares (mathematica, matlab,...) makes the work easily? In any case I found this Appendix difficult to follow and I would ask the authors to smooth it out.*

Authors: *We have dropped the details of the fixed point search and we have considerably shortened Appendix A to make it easier to follow.*

Ref. #1: *I think it can be helpful for the discussion if you show the full set of advection-reaction- diffusion equations with all the terms and not in different pieces (Eq (3-4), and all the discussion below). In particular, the way the authors address the air-sea exchange flux term was unclear for me.*

Authors: *We have improved the way we present the model equations in the new version of the manuscript. All terms pertaining to dissolved $O_2$, P and Z are now shown as one fully-developed differential equation for each species:*

"In the coupled model, the time evolution of DO, P and Z concentrations are given by, respectively:

$$\partial[O_2]/\partial t + u\partial[O_2]/\partial x + v\partial[O_2]/\partial y + w\partial[O_2]/\partial z = k_H (\partial^2[O_2]/\partial x^2 + \partial^2[O_2]/\partial y^2) + \partial/\partial z(<[O_2]'w'> - k_V\partial[O_2]/\partial z) + Va ([O_2] - [O_2]_{sat}) + AJ(z)f([O_2])[P] - u_r([O_2],[P]) - v_r([O_2],[Z]) - m[O_2], \quad (1)$$

[revised manuscript text omitted]

**Insertions:**

Upwelling systems are thus especially sensitive to episodic and long-term changes in DO levels with deleterious effects on marine life and human activities such as fisheries (Grantham et al., 2004; Hales et al., 2006; Paulmier and Ruiz-Pino, 2009; McClatchie et al. 2010; Roegner et al., 2011).

DO levels in these subsurface waters are thought to diminish during the upwelling season due to strong remineralization of sinking organic matter as well as to mixing with poorly-ventilated waters of subtropical origins (Castro et al., 2006). Overall, DO appears to partly control the remineralization of dissolved organic carbon in the IPUS and more generally in key water masses of the North Atlantic (Alvarez-Salgado et al. 2013).

In the Iberian basin, the Eastern North Atlantic Central Water (ENACW) mass of subtropical origin, whose archetypal concentrations of DO are initially low, would be ventilated by eddy-induced mixing with more oxygenated ENACW of subpolar origin (Pèrez et al., 2001).

Moreover, diatoms growth is thought to be positively correlated with the upwelling-driven nitrate inputs (Sarthou et al., 2005); in contrast, the growth of other planktonic groups can be insensitive to and even

limited by newly-upwelled nitrates (Mahaffey, 2005), especially when colimitation prevails or in the absence of necessary micro-nutrients.

The low-complexity $O_2PZ$ model was directly implemented into the CROCO model following one of the already-embedded biogeochemical modules; it allows maintaining the full capabilities and versatility of this community code while gaining in computing efficiency (allowing extensive sensitivity studies and the consideration of submesoscale dynamics in oxygen cycling) as compared to the more complex biogeochemical modules already available within CROCO.

In the coupled model, the time evolution of DO, P and Z concentrations are given by, respectively:

$$\partial[O_2]/\partial t + u\partial[O_2]/\partial x + v\partial[O_2]/\partial y + w\partial[O_2]/\partial z = k_H (\partial^2[O_2]/\partial x^2 + \partial^2[O_2]/\partial y^2) + \partial/\partial z(<[O_2]'w'> - k_V\partial[O_2]/\partial z) + Va ([O_2] - [O_2]_{sat}) + AJ(z)f([O_2])[P] - u_r([O_2],[P]) - v_r([O_2],[Z]) - m[O_2],$$
(1)

$$\partial[P]/\partial t + u\partial[P]/\partial x + v\partial[P]/\partial y + w\partial[P]/\partial z = k_H (\partial^2[P]/\partial x^2 + \partial^2[P]/\partial y^2) + \partial/\partial z(<[P]'w'> - k_V\partial[P]/\partial z) + l(N)g([O_2],[P]) - e([P],[Z]) - \sigma[P],$$
(2)

$$\partial[Z]/\partial t + u\partial[Z]/\partial x + v\partial[Z]/\partial y + w\partial[Z]/\partial z = k_H (\partial^2[Z]/\partial x^2 + \partial^2[Z]/\partial y)^2) + \partial/\partial z(<[Z]'w'> - k_V \partial[Z]/\partial z) + \kappa([O_2])e([P],[Z]) - \mu[Z],$$
(3)

where $(u,v,w)$ is the 3D velocity field, $k_H$ and $k_V$ are the horizontal and vertical tracer diffusivities, and $<...>$ are the vertical turbulent fluxes. The biogeochemical model $O_2PZ$ (SP2015) defines the source terms of DO, P and Z in the coupled model. It simulates seven biogeochemical reactions between the three compartments (Fig. 2) as follows: in the DO equation (Eq. 1),

On top of the oxygen that is produced and consumed within marine ecosystems, a considerable part of dissolved oxygen eventually becomes gaseous and is then exchanged at the ocean-atmosphere interface, contributing to the estimations that more than one half of atmospheric oxygen is produced in the ocean (Harris, 1986).The remaining term of Eq. (2) is the air-sea flux of $O_2$, which is the product of the gas transfer velocity velocity $Va$, that depends on the square of the wind speed and on the Schmidt number (Wanninkhof, 1992), with the difference of the oceanic DO concentration and the solubility of $O_2$ in seawater.

This term is included here to circumvent the absence of nutrient limitation in the original $O_2PZ$ model of SP2015. We chose to add a parametrization instead of a new equation for the nutrients because this last option would necessarily increase the complexity of the model.

The cross-shore wind component is zero in all cases. Despite the fact that "true" wind patterns observed over Western Iberia are undoubtedly more variable (both in terms of direction and magnitude) than our idealized forcing inspired from a specific campaign, it provides the quickest development of a clear upwelling front generating small-scale instabilities. Note that sensitivity tests were performed (not shown) and revealed that the model responses are qualitatively similar among different wind strengths, given that the magnitude is high enough to ensure the development of the unstable upwelling front.

We compared the full ranges of densities $\rho$, DO concentrations $[O_2]$, and chlorophyll-a concentrations [Chl-a] of the reference simulation to all compiled measurements collected during the MOUTON campaign. Given the low-complexity of the biogechemical model and the highly idealized physical setting, a fully quantitative comparison is out of scope, however we confirm that the coupled model reproduces qualitatively well the $\rho$-$[O_2]$ and $\rho$-[Chl-a] relationships obtained from in-situ observations.

In a qualitative sense our idealized model results are similar to more realistic model studies such as , Gutknecht *et al.* (2013) which in their modelling study of the Benguela upwelling region found a low $[O_2]$ plume for the climatological month of December at the shelf edge, and to observational studies such as Hales *et al.* (2006), which for the Oregon coast, measured $[O_2]$ of 70-110 mmol $m^{-3}$ in upwelled water at the shelf break, about 200 mmol $m^{-3}$ less than at the surface, which is the same range of the vertical gradient simulated here.

The low-complexity $O_2PZ$ model accounts for oxygen production by photosynthesis and consumption by both respiration and remineralization but its simplicity does not allow to include other oceanographic

processes that could influence oxygen cycling over shallow shelves, such as for instance the complex benthic-pelagic coupled processes (Capet et al., 2016).

compare

qualitatively

In permanent Oxygen Minimum Zones (OMZs), changes in physical forcing (e.g. wind regime) and/or biological characteristics (e.g. size structure or functional types) can have dire implications for marine ecosystems and fisheries management. Indeed, the measured expansion of the Tropical Atlantic and Pacific OMZs (Stramma et al., 2008) implies a shallower upper boundary of the low $O_2$ core and an increased sensitivity of oxygen levels over the continental shelf to upwelling winds, which are indeed predicted to intensify with climate change (Sydeman et al., 2014). Future work could be focused on understanding this high sensitivity and on anticipating how it would affect the stability of the coupled-system in a changing climate.

This results in a

that

Álvarez-Salgado, X.A., Nieto-Cid, M., Álvarez, M., Pérez, F.F., Morin, P., Mercier, H.: New insights on the mineralization of dissolved organic matter in central, intermediate, and deep water masses of the northeast North Atlantic. Limnol. Oceanogr. 58, 681–696. doi.org/10.4319/lo.2013.58.2.0681, 2013.

Capet, A., Meysman, F.J.R., Akoumianaki, I., Soetaert, K., Grégoire, M.: Integrating sediment biogeochemistry into 3D oceanic models: A study of benthic-pelagic coupling in the Black Sea. Ocean Modelling 101, 83–100. doi.org/10.1016/j.ocemod.2016.03.006, 2016.

Castro, C.G., Nieto-Cid, M., Álvarez-Salgado, X.A., Pérez, F.F.: Local remineralization patterns in the mesopelagic zone of the Eastern North Atlantic, off the NW Iberian Peninsula. Deep Sea Research Part I: Oceanographic Research Papers 53, 1925–1940. doi.org/10.1016/j.dsr.2006.09.002, 2006.

Laffoley, D. and Baxter, J.,2019, Ocean deoxygenation : everyone's problem : causes, impacts, consequences and solutions, IUCN Report, Gland, Switzerland, 562 pages, doi.org/10.2305/IUCN.CH.2019.13.en

Pérez, F.F., Castro, C.G., Álvarez–Salgado, X.A., Ríos, A.F.: Coupling between the Iberian basin — scale circulation and the Portugal boundary current system: a chemical study. Deep Sea Research Part I: Oceanographic Research Papers 48, 1519–1533. doi.org/10.1016/S0967-0637(00)00101-1, 2001.

Sarthou, G., Timmermans, K.R., Blain, S., Tréguer, P.: Growth physiology and fate of diatoms in the ocean: a review. Journal of Sea Research, Iron Resources and Oceanic Nutrients - Advancement of Global Environmental Simulations 53, 25–42. doi.org/10.1016/j.seares.2004.01.007, 2005.

Stramma, L., Johnson, G.C., Sprintall, J., Mohrholz, V.: Expanding Oxygen-Minimum Zones in the Tropical Oceans. Science 320, 655–658. doi.org/10.1126/science.1153847, 2008.

Sydeman, W.J., García-Reyes, M., Schoeman, D.S., Rykaczewski, R.R., Thompson, S.A., Black, B.A., Bograd, S.J.: Climate change and wind intensification in coastal upwelling ecosystems. Science 345, 77–80. doi.org/10.1126/science.1251635, 2014.

Wanninkhof, R.: Relationship between wind speed and gas exchange over the ocean, Journal of Geophysical Research, 97(C5), 7373, doi:10.1029/92JC00188, 1992.

$$\partial[P]/\partial t + u\partial[P]/\partial x + v\partial[P]/\partial y + w\partial[P]/\partial z = k_H (\partial^2[P]/\partial x^2 + \partial^2[P]/\partial y^2) + \partial/\partial z(<[P]'w'> -$$

$$k_V\partial[P]/\partial z) + l(N)g([O_2],[P]) - e([P],[Z]) - \sigma[P], \tag{2}$$

$$\partial[Z]/\partial t + u\partial[Z]/\partial x + v\partial[Z]/\partial y + w\partial[Z]/\partial z = k_H (\partial^2[Z]/\partial x^2 + \partial^2[Z]/\partial y)^2) + \partial/\partial z(<[Z]'w'> - k_V$$

$$\partial[Z]/\partial z) + \kappa([O_2])e([P],[Z]) - \mu[Z], \tag{3}$$

where $(u,v,w)$ is the 3D velocity field, $k_H$ and $k_V$ are the horizontal and vertical tracer diffusivities, and $<...>$ are the vertical turbulent fluxes."

*We have also rewritten the text and developed our explanations to better describe the Air-Sea exchange term:*

"On top of the oxygen that is produced and consumed within marine ecosystems, a considerable part of dissolved oxygen eventually becomes gaseous and is then exchanged at the ocean-atmosphere interface, contributing to the estimations that more than one half of atmospheric oxygen is produced in the ocean (Harris, 1986).The remaining term of Eq. (2) is the air-sea flux of $O_2$, which is the product of the gas transfer velocity velocity $Va$, that depends on the square of the wind speed and on the Schmidt number (Wanninkhof, 1992), with the difference of the oceanic DO concentration and  the solubility of $O_2$ in seawater."

Ref. #1: *After reading several times, I do not understand Fig. 6e. Please clarify its meaning. Is it really helpful?*

Authors : *Figure 6e displays the alongshore averaged concentrations of $O_2$ for each binned value of the Okubo-Weiss criterion ($\gamma$) . It is computed from the data of figure 6d by averaging the values in the map "horizontally", that is along the x-axis for a constant $\gamma$ level. It emphasizes the result that, on average, there are higher $O_2$ concentrations in the rotation-dominated regions ($\gamma < 0$) as compared against strain-dominated regions ($\gamma > 0$); this information is not readily observable in figure 6d.*

Ref. #1: *In the discussion/conclusions section I missed some discussion on the relevance of the results of this work on permanent marine oxygen minimum zones*

Authors : *Thank you for this pertinent suggestion; we have added a few words about this topic in the discussion section:*

"In permanent Oxygen Minimum Zones (OMZs), changes in physical forcing (e.g. wind regime) and/or biological characteristics (e.g. size structure or functional types) can have dire implications for marine ecosystems and fisheries management. Indeed, the measured expansion of the Tropical Atlantic and Pacific OMZs (Stramma et al., 2008) implies a shallower upper boundary of the low $O_2$ core and an increased sensitivity of oxygen levels over the continental shelf to upwelling winds, which are indeed predicted to intensify with climate change (Sydeman et al., 2014). Future work could be focused on understanding this high sensitivity and on anticipating how it would affect the stability of the coupled-system in a changing climate."

**Authors' reply to Referee #2**

We thank the Referee for his/her comments that helped to greatly improve our manuscript. Below, we provide a point-by-point reply to all referee's comments.

Ref. #2: *While I generally agree with the comments of Referee #1, in terms of the quality of the manuscript, and its relevance of publication, there are a number of points that I will refer to, which should be addressed before the final publication. On the one hand, the con- figuration used is highly idealized, and the practical application to the Iberian system is not straightforward. Provided the simplifications that were done related the idealized configuration, and the theoretical approach, (quite difficult to read to oceanographers not specialized in theoretical oceanography) I have some doubts about the practical applicability of the results, and the interest that it may have for the oceanographic community of the region, devoted to the study of biogeochemical processes, (constituted mainly by chemical oceanographers).*

Authors: *We acknowledge the concerns of Referee 2 concerning the future applicability of our results due to the theoretical character of our study. This contribution is indeed, to our knowledge, the first and unique attempt to develop a low complexity ocean coupled model ("NPZD-type", which has been proven very useful for the Oceanographic community) centered on oxygen by building upon recent mathematical modelling. To strengthen the utility and the potential applicability of our results, we have rewritten several parts of the manuscript (shown below in the replies) to better summarize our results and make it easier for the oceanographic community to follow the theoretical description of our model (please check the revised appendix A).*

Ref. #2: *1 - Introduction: Paragraph 1. The Iberian region is not an example of a region of marine hypoxia, and that should be referred to in the manuscript. References to de- oxygenation (line 40) do not make much sense in the scope of this manuscript.*

Authors: *Following your recommendations, we have modified the Introduction section by removing the references to marine hypoxia and deoxygenation and by introducing additional references as suggested by the next comment. Both first paragraphs of the Introduction have been rewritten accordingly:*

"Declining dissolved oxygen (DO) in the world ocean and coastal realm impacts all marine life, from microbes to higher trophic levels (Breitburg et al., 2018), with consequences ranging from ecological adaptations and shifts (Gilly et al., 2013), changes of biogeochemical activity (Wright et al., 2012) to mass mortality events (Diaz and Rosenberg, 2008) and biodiversity restructuring (Vaquer-Sunyer and Duarte, 2008).
Coastal waters are generally eutrophic and characterized by substantial planktonic productivity at the surface which favours oxygen consumption through remineralization of sinking organic matter, leading to low levels of DO in subsurface and near-bottom waters. Highly productive surface waters are typical of coastal upwelling regions and sustain socio-economically important ecosystems. Upwelling systems are thus especially sensitive to episodic and long-term changes in DO levels with deleterious effects on marine life and human activities such as fisheries (Grantham et al., 2004; Hales et al., 2006; Paulmier and Ruiz-Pino, 2009; McClatchie et al. 2010; Roegner et al., 2011)."

Ref. #2: *2 - The authors barely refer to the observational literature related to the oxygen cycle in the Iberian region, limiting themselves to cite Rossi's articles (2010, 2013), which consist of a single campaign, not particularly devoted to that subject. There is some observational literature including the oxygen cycle that I think should be cited, as it is relevant. (Hint: Alvarez-Salgado, X.A, Perez, F.F, Castro, C.G. , all of them in IIM-CSIC-Vigo, Spain) On the other hand there are oceanographic processes not solved by the model that*

[revised manuscript text omitted]

**Insertions:**

Upwelling systems are thus especially sensitive to episodic and long-term changes in DO levels with deleterious effects on marine life and human activities such as fisheries (Grantham et al., 2004; Hales et al., 2006; Paulmier and Ruiz-Pino, 2009; McClatchie et al. 2010; Roegner et al., 2011).

DO levels in these subsurface waters are thought to diminish during the upwelling season due to strong remineralization of sinking organic matter as well as to mixing with poorly-ventilated waters of subtropical origins (Castro et al., 2006). Overall, DO appears to partly control the remineralization of dissolved organic carbon in the IPUS and more generally in key water masses of the North Atlantic (Alvarez-Salgado et al. 2013).

In the Iberian basin, the Eastern North Atlantic Central Water (ENACW) mass of subtropical origin, whose archetypal concentrations of DO are initially low, would be ventilated by eddy-induced mixing with more oxygenated ENACW of subpolar origin (Pèrez et al., 2001).

Moreover, diatoms growth is thought to be positively correlated with the upwelling-driven nitrate inputs (Sarthou et al., 2005); in contrast, the growth of other planktonic groups can be insensitive to and even

limited by newly-upwelled nitrates (Mahaffey, 2005), especially when colimitation prevails or in the absence of necessary micro-nutrients.

The low-complexity $O_2PZ$ model was directly implemented into the CROCO model following one of the already-embedded biogeochemical modules; it allows maintaining the full capabilities and versatility of this community code while gaining in computing efficiency (allowing extensive sensitivity studies and the consideration of submesoscale dynamics in oxygen cycling) as compared to the more complex biogeochemical modules already available within CROCO.

In the coupled model, the time evolution of DO, P and Z concentrations are given by, respectively:

$$\partial[O_2]/\partial t + u\partial[O_2]/\partial x + v\partial[O_2]/\partial y + w\partial[O_2]/\partial z = k_H (\partial^2[O_2]/\partial x^2 + \partial^2[O_2]/\partial y^2) + \partial/\partial z(<[O_2]'w'> - k_V\partial[O_2]/\partial z) + Va ([O_2] - [O_2]_{sat}) + AJ(z)f([O_2])[P] - u_r([O_2],[P]) - v_r([O_2],[Z]) - m[O_2],$$
(1)

$$\partial[P]/\partial t + u\partial[P]/\partial x + v\partial[P]/\partial y + w\partial[P]/\partial z = k_H (\partial^2[P]/\partial x^2 + \partial^2[P]/\partial y^2) + \partial/\partial z(<[P]'w'> - k_V\partial[P]/\partial z) + l(N)g([O_2],[P]) - e([P],[Z]) - \sigma[P],$$
(2)

$$\partial[Z]/\partial t + u\partial[Z]/\partial x + v\partial[Z]/\partial y + w\partial[Z]/\partial z = k_H (\partial^2[Z]/\partial x^2 + \partial^2[Z]/\partial y)^2) + \partial/\partial z(<[Z]'w'> - k_V \partial[Z]/\partial z) + \kappa([O_2])e([P],[Z]) - \mu[Z],$$
(3)

where $(u,v,w)$ is the 3D velocity field, $k_H$ and $k_V$ are the horizontal and vertical tracer diffusivities, and $<...>$ are the vertical turbulent fluxes. The biogeochemical model $O_2PZ$ (SP2015) defines the source terms of DO, P and Z in the coupled model. It simulates seven biogeochemical reactions between the three compartments (Fig. 2) as follows: in the DO equation (Eq. 1),

On top of the oxygen that is produced and consumed within marine ecosystems, a considerable part of dissolved oxygen eventually becomes gaseous and is then exchanged at the ocean-atmosphere interface, contributing to the estimations that more than one half of atmospheric oxygen is produced in the ocean (Harris, 1986).The remaining term of Eq. (2) is the air-sea flux of $O_2$, which is the product of the gas transfer velocity velocity $Va$, that depends on the square of the wind speed and on the Schmidt number (Wanninkhof, 1992), with the difference of the oceanic DO concentration and the solubility of $O_2$ in seawater.

This term is included here to circumvent the absence of nutrient limitation in the original $O_2PZ$ model of SP2015. We chose to add a parametrization instead of a new equation for the nutrients because this last option would necessarily increase the complexity of the model.

The cross-shore wind component is zero in all cases. Despite the fact that "true" wind patterns observed over Western Iberia are undoubtedly more variable (both in terms of direction and magnitude) than our idealized forcing inspired from a specific campaign, it provides the quickest development of a clear upwelling front generating small-scale instabilities. Note that sensitivity tests were performed (not shown) and revealed that the model responses are qualitatively similar among different wind strengths, given that the magnitude is high enough to ensure the development of the unstable upwelling front.

We compared the full ranges of densities $\rho$, DO concentrations $[O_2]$, and chlorophyll-a concentrations [Chl-a] of the reference simulation to all compiled measurements collected during the MOUTON campaign. Given the low-complexity of the biogechemical model and the highly idealized physical setting, a fully quantitative comparison is out of scope, however we confirm that the coupled model reproduces qualitatively well the $\rho$-$[O_2]$ and $\rho$-[Chl-a] relationships obtained from in-situ observations.

In a qualitative sense our idealized model results are similar to more realistic model studies such as , Gutknecht *et al.* (2013) which in their modelling study of the Benguela upwelling region found a low $[O_2]$ plume for the climatological month of December at the shelf edge, and to observational studies such as Hales *et al.* (2006), which for the Oregon coast, measured $[O_2]$ of 70-110 mmol m$^{-3}$ in upwelled water at the shelf break, about 200 mmol m$^{-3}$ less than at the surface, which is the same range of the vertical gradient simulated here.

The low-complexity $O_2PZ$ model accounts for oxygen production by photosynthesis and consumption by both respiration and remineralization but its simplicity does not allow to include other oceanographic

processes that could influence oxygen cycling over shallow shelves, such as for instance the complex benthic-pelagic coupled processes (Capet et al., 2016).

compare

qualitatively

In permanent Oxygen Minimum Zones (OMZs), changes in physical forcing (e.g. wind regime) and/or biological characteristics (e.g. size structure or functional types) can have dire implications for marine ecosystems and fisheries management. Indeed, the measured expansion of the Tropical Atlantic and Pacific OMZs (Stramma et al., 2008) implies a shallower upper boundary of the low $O_2$ core and an increased sensitivity of oxygen levels over the continental shelf to upwelling winds, which are indeed predicted to intensify with climate change (Sydeman et al., 2014). Future work could be focused on understanding this high sensitivity and on anticipating how it would affect the stability of the coupled-system in a changing climate.

This results in a

that

Álvarez-Salgado, X.A., Nieto-Cid, M., Álvarez, M., Pérez, F.F., Morin, P., Mercier, H.: New insights on the mineralization of dissolved organic matter in central, intermediate, and deep water masses of the northeast North Atlantic. Limnol. Oceanogr. 58, 681–696. doi.org/10.4319/lo.2013.58.2.0681, 2013.

Capet, A., Meysman, F.J.R., Akoumianaki, I., Soetaert, K., Grégoire, M.: Integrating sediment biogeochemistry into 3D oceanic models: A study of benthic-pelagic coupling in the Black Sea. Ocean Modelling 101, 83–100. doi.org/10.1016/j.ocemod.2016.03.006, 2016.

Castro, C.G., Nieto-Cid, M., Álvarez-Salgado, X.A., Pérez, F.F.: Local remineralization patterns in the mesopelagic zone of the Eastern North Atlantic, off the NW Iberian Peninsula. Deep Sea Research Part I: Oceanographic Research Papers 53, 1925–1940. doi.org/10.1016/j.dsr.2006.09.002, 2006.

Laffoley, D. and Baxter, J.,2019, Ocean deoxygenation : everyone's problem : causes, impacts, consequences and solutions, IUCN Report, Gland, Switzerland, 562 pages, doi.org/10.2305/IUCN.CH.2019.13.en

Pérez, F.F., Castro, C.G., Álvarez–Salgado, X.A., Ríos, A.F.: Coupling between the Iberian basin — scale circulation and the Portugal boundary current system: a chemical study. Deep Sea Research Part I: Oceanographic Research Papers 48, 1519–1533. doi.org/10.1016/S0967-0637(00)00101-1, 2001.

Sarthou, G., Timmermans, K.R., Blain, S., Tréguer, P.: Growth physiology and fate of diatoms in the ocean: a review. Journal of Sea Research, Iron Resources and Oceanic Nutrients - Advancement of Global Environmental Simulations 53, 25–42. doi.org/10.1016/j.seares.2004.01.007, 2005.

Stramma, L., Johnson, G.C., Sprintall, J., Mohrholz, V.: Expanding Oxygen-Minimum Zones in the Tropical Oceans. Science 320, 655–658. doi.org/10.1126/science.1153847, 2008.

Sydeman, W.J., García-Reyes, M., Schoeman, D.S., Rykaczewski, R.R., Thompson, S.A., Black, B.A., Bograd, S.J.: Climate change and wind intensification in coastal upwelling ecosystems. Science 345, 77–80. doi.org/10.1126/science.1251635, 2014.

Wanninkhof, R.: Relationship between wind speed and gas exchange over the ocean, Journal of Geophysical Research, 97(C5), 7373, doi:10.1029/92JC00188, 1992.

**Effects of upwelling duration and phytoplankton growth regime on dissolved oxygen levels in an idealized Iberian Peninsula upwelling system**

J. H. Bettencourt[1*], V. Rossi[2], L. Renault[1], P. Haynes[3], Y. Morel[1], V. Garçon[1]

[1]LEGOS, University of Toulouse, CNES, CNRS, IRD, UPS, Toulouse 31400, France

[2]MIO (UM 110, UMR 7294), CNRS, Aix-Marseille Univ., Univ. Toulon, IRD, 13288, Marseille, France

[3]Department of Applied Mathematics and Theoretical Physics, University of Cambridge, England

*Correspondence to:
J. H. Bettencourt (joao.bettencourt@tecnico.ulisboa.pt)
Marine Environment Group
CENTEC - Centre for Marine Technology and Ocean Engineering
Instituto Superior Técnico (Pavilhão Central)
1049-001 Lisboa
Portugal

**Abstract.** We apply a coupled modelling system composed of a state-of-the-art hydrodynamical model and a low complexity biogeochemical model to an idealized Iberian Peninsula upwelling system to identify the main drivers of dissolved oxygen variability and to study its response to changes in the duration of the upwelling season and in phytoplankton growth regime. We find that the export of oxygenated waters by upwelling front turbulence is a major sink for nearshore dissolved oxygen. In our simulations of summer upwelling, when phytoplankton population is generally dominated by diatoms whose growth is largely enhanced by nutrient input, net primary production and air-sea exchange compensate dissolved oxygen depletion by offshore export over the shelf. A shorter upwelling duration causes relaxation of upwelling winds and a decrease in offshore export, resulting in a slight increase of net dissolved oxygen enrichment in the coastal region as compared to longer upwelling durations. When phytoplankton is dominated by groups less sensitive to nutrient inputs, growth rates decrease and the coastal region becomes net heterotrophic. Together with the physical sink, this lowers the net oxygenation rate of coastal waters, that remains positive only because of air-sea exchanges. These findings help disentangling the physical and biogeochemical controls of dissolved oxygen in upwelling systems and, together with projections of increased duration of upwelling seasons and phytoplankton community changes, suggest that the Iberian coastal upwelling region may become more vulnerable to hypoxia and deoxygenation.

**1 Introduction**

[Deleted Text] Declining dissolved oxygen (DO) in the world ocean and coastal realm impacts all marine life, from microbes to higher trophic levels (Breitburg et al., 2018), with consequences ranging from ecological adaptations and shifts (Gilly et al., 2013), changes of biogeochemical activity (Wright et al., 2012) to mass mortality events (Diaz and Rosenberg, 2008) and biodiversity restructuring (Vaquer-Sunyer and Duarte, 2008).

Coastal waters are generally eutrophic and characterized by substantial planktonic productivity at the surface which favours oxygen consumption through remineralization of sinking organic matter, leading to low levels of DO in subsurface and near-bottom waters. Highly productive surface waters are typical of coastal upwelling regions and sustain socio-economically important ecosystems.

 [Inserted text]Upwelling systems are thus especially sensitive to episodic and long-term changes in DO levels with deleterious effects on marine life and human activities such as fisheries (Grantham et al., 2004; Hales et al., 2006; Paulmier and Ruiz-Pino, 2009; McClatchie et al. 2010; Roegner et al., 2011).

The western Iberian Peninsula Upwelling System (IPUS) is the northern branch of the Canary Upwelling System where the intra-annual variability of alongshore winds produce a seasonal upwelling/downwelling cycle (Wooster et al., 1976). In the summer and early fall, the Azores High migrates northward, causing a poleward alongshore wind that forces offshore Ekman transport of surface waters and upwelling of subsurface waters, while downwelling prevails the rest of the year. Long-term studies of the seasonal upwelling in the IPUS have pointed to a weakening of upwelling winds over multidecadal time scales (Sousa et al., 2017), but simulations of future climate indicate an enhancement of the upwelling due to poleward migration of the Azores High and a lengthening of the upwelling season  (Miranda et al., 2013; Rykaczewski et al., 2015; Sousa et al., 2017).

During the upwelling season, hydrographic and biogeochemical variability is primarily determined by the wind forcing that controls the inflow of offshore deep water masses onto the shelf (Alvarez-Salgado *et al.*, 1993). DO levels in these subsurface waters are thought to diminish during the upwelling season due to strong remineralization of sinking organic matter as well as to mixing with poorly-ventilated waters of subtropical origins (Castro et al., 2006). Overall, DO appears to partly control the remineralization of dissolved organic carbon in the IPUS and more generally in key water masses of the North Atlantic (Alvarez-Salgado et al. 2013).

[revised manuscript text omitted]

 (1)

 (2)

 (3)

 (4)

In the coupled model, the time evolution of DO, P and Z concentrations are given by, respectively:

$\partial[O_2]/\partial t + u\partial[O_2]/\partial x + v\partial[O_2]/\partial y + w\partial[O_2]/\partial z = k_H (\partial^2[O_2]/\partial x^2 + \partial^2[O_2]/\partial y^2) + \partial/\partial z(<[O_2]'w'>-k_V\partial[O_2]/\partial z) + Va ([O_2] - [O_2]_{sat}) + AJ(z)f([O_2])[P]-u_r([O_2],[P])-v_r([O_2],[Z])-m[O_2]$, (1)

$\partial[P]/\partial t + u\partial[P]/\partial x + v\partial[P]/\partial y + w\partial[P]/\partial z = k_H (\partial^2[P]/\partial x^2 + \partial^2[P]/\partial y^2) + \partial/\partial z(<[P]'w'>- k_V\partial[P]/\partial z) + l(N)g([O_2],[P])-e([P],[Z])-\sigma[P]$, (2)

$\partial[Z]/\partial t + u\partial[Z]/\partial x + v\partial[Z]/\partial y + w\partial[Z]/\partial z = k_H (\partial^2[Z]/\partial x^2 + \partial^2[Z]/\partial y)^2) + \partial/\partial z(<[Z]'w'>- k_V \partial[Z]/\partial z) + \kappa([O_2])e([P],[Z])-\mu[Z]$, (3)

where (u,v,w) is the 3D velocity field, $k_H$ and $k_V$ are the horizontal and vertical tracer diffusivities, and <...> are the vertical turbulent fluxes. The biogeochemical model O$_2$PZ (SP2015) defines the source terms of DO, P and Z in the coupled model. It simulates seven biogeochemical reactions between the three compartments (Fig. 2) as follows: in the DO equation (Eq. 1), the term $AJ(z)f([O_2])$ is the rate of O$_2$ production by photosynthesis and is modelled by a Monod parametrization $f([O_2])=c_0/([O_2]+c_0)$, where $c_0$ is a half-saturation constant. The term $J(z) = 1 – exp(\alpha PAR(z)/A)$ represents the light dependency of photosynthesis, with $\alpha = 30$ and the photosynthetically available radiation $PAR(z) = PAR(0) exp(-k_w z)$, with $PAR(0) = 355.19$ W/m$^2$ and $k_w = 0.04$ m$^{-1}$.

[revised manuscript text omitted]

We compared the full ranges of densities ρ, DO concentrations [O2], and chlorophyll-a concentrations [Chl-a] of the reference simulation to all compiled measurements collected during the MOUTON campaign. Given the low-complexity of the biogechemical model and the highly idealized physical setting, a fully quantitative comparison is out of scope, however we confirm that the coupled model reproduces qualitatively well the ρ-[O2] and ρ-[Chl-a] relationships obtained from in-situ observations.

**3. Results and discussion**

**3.1 Dissolved Oxygen in the Idealized IPUS**

Our coupled model reproduces well the upwelling circulation and the typical biological responses. Indeed, the upwelling of nutrient-rich waters induced by the mean wind-driven circulation promotes the growth of phytoplankton, leading to increased oxygen production by photosynthesis and the subsequent enrichment of oxygen in shelf waters (x < 80 km, Fig. 5(a)), as compared with the lower [O2] found in offshore waters (x > 80 km). It also shows a low [O2] cell (~ 50 mmol m$^{-3}$), at the subsurface over the outer-shelf (centred at 40 km and 60 m depth) because of the upwelling of low O2 waters, consistent with the shelf low [O2] cell (< 200 µmol kg-1) measured during the MOUTON cruise (e.g. Rossi et al. 2013). ~~Similarly, Gutknecht et al. (2013) in their modelling study of the Benguela upwelling region found a low [O2] plume for the climatological month of December at the shelf edge. For the Oregon coast, Hales et al. (2006) measured [O2] of 70-110 mmol m$^{-3}$ in upwelled water at the shelf break, about 200 mmol m$^{-3}$ less than at the surface, which is the same range of the vertical gradient simulated here.~~ In a qualitative sense our idealized model results are similar to more realistic model studies such as , Gutknecht et al. (2013) which in their modelling study of the Benguela upwelling region found a low [O2] plume for the climatological month of December at the shelf edge, and to observational studies such as Hales et al. (2006), which for the Oregon coast, measured [O2] 
[revised manuscript text omitted]
. The low-complexity O$_2$PZ model accounts for oxygen production by photosynthesis and consumption by both respiration and remineralization but its simplicity does not allow to include other oceanographic processes that could influence oxygen cycling over shallow shelves, such as for instance the complex benthic-pelagic coupled processes (Capet et al., 2016). While we used data from the Iberian Peninsula upwelling to initialize and  compare the range of values simulated by our model, our idealized configuration allows to draw general conclusions about the mechanisms governing the dissolved oxygen levels over the continental shelf. When compared to measurements, our model reproduces qualitatively the O$_2$-density relationship as well as upwelling of oxygen-poor waters onto the shelf and the offshore transport of oxygen due to filaments and vortices. While the addition of air-sea exchange processes as well as our novel nutrient/density parametrization have probably contributed largely to simulate such realistic outputs despite the simple biogeochemical formulation, the coupled system could be further improved. One probably important task, that we keep for future studies, is to include realist subsurface O$_2$ inputs from oceanic currents, as it has been shown to control largely O$_2$ dynamics over the shelf (Montes et al., 2014; Reboreda et al., 2015).

Although there is a clear separation of coastal waters with high [O$_2$] and offshore waters with low [O$_2$], the turbulent lateral transport of high [O$_2$] water disrupts this picture, as oxygen-rich upwelling filaments and vortices generated at the front and then travel offshore. This creates a sink of dissolved oxygen on the shelf that must be compensated by photosynthetic production and air-sea fluxes. In our reference simulation, with repeated upwelling wind cycle, the net [O$_2$] rate of change was positive, owing to the predominance of the photosynthetic oxygen production and air-sea fluxes over the lateral transport of oxygen, that acts to remove it from the shelf region. When considering a shorter upwelling season (the ECS case with four 10-day upwelling wind cycles followed by wind cessation) reinforces somewhat the oxygen enrichment trend. When phytoplankton community is dominated by groups that show nutrient limitation or neutral growth (LCC and NCC cases), the biological source weakens and the shelf becomes net heterotrophic. The physical sink is also affected, producing a weaker loss of dissolved oxygen in the coastal upwelling. Air-sea exchange then maintains the (much lower) net oxygen enrichment trend Our findings that the oxygen inventory net rate of change is inversely proportional to the duration of upwelling season is consistent with other studies (Adams et al., 2013; Siedlecki et al., 2015; Zhang et al., 2018) and, as the upwelling season duration in the IPUS is thought to increase in the future (Miranda et al., 2013; Wang et al., 2015), it suggests increased vulnerability of coastal regions to hypoxia and long-term deoxygenation. The result of lower phytoplankton growth rates driving the shelf region from autotrophy to heterotrophy also points to an increased risk of loss of dissolved oxygen if trends for community shifts towards smaller or less efficient phytoplankton (Gomes et al., 2014; Gregg et al., 2017; Head and Pepin, 2010; Zhai et al., 2013) are specifically confirmed for the IPUS. In permanent Oxygen Minimum Zones (OMZs), changes in

physical forcing (e.g. wind regime) and/or biological characteristics (e.g. size structure or functional types) can have dire implications for marine ecosystems and fisheries management. Indeed, the measured expansion of the Tropical Atlantic and Pacific OMZs (Stramma et al., 2008) implies a shallower upper boundary of the low $O_2$ core and an increased sensitivity of oxygen levels over the continental shelf to upwelling winds, which are indeed predicted to intensify with climate change (Sydeman et al., 2014). Future work could be focused on understanding this high sensitivity and on anticipating how it would affect the stability of the coupled-system in a changing climate.

**Data availability**

[revised manuscript text omitted]

Álvarez-Salgado, X.A., Nieto-Cid, M., Álvarez, M., Pérez, F.F., Morin, P., Mercier, H.: New insights on the mineralization of dissolved organic matter in central, intermediate, and deep water masses of the northeast North Atlantic. Limnol. Oceanogr. 58, 681–696. doi.org/10.4319/lo.2013.58.2.0681, 2013.

Behrenfeld, M. J., Halsey Kimberly H and Milligan Allen J: Evolved physiological responses of phytoplankton to their integrated growth environment, Philosophical Transactions of the Royal Society B: Biological Sciences, 363(1504), 2687–2703, doi:10.1098/rstb.2008.0019, 2008.

Bettencourt, J. H., López, C., Hernández-García, E., Montes, I., Sudre, J., Dewitte, B., Paulmier, A. and Garçon, V.: Boundaries of the Peruvian oxygen minimum zone shaped by coherent mesoscale dynamics, Nature Geoscience, 8(12), 937–940, doi:10.1038/ngeo2570, 2015.

Bettencourt, J. H., Rossi, V., Hernández-García, E., Marta-Almeida, M. and López, C.: Characterization of the Structure and Cross-Shore Transport Properties of a Coastal Upwelling Filament Using Three-Dimensional Finite-Size Lyapunov Exponents, Journal of Geophysical Research: Oceans, 122(9), 7433–7448, doi:10.1002/2017JC012700, 2017.

Boyer, T. P., Antonov, J. I., Baranova, O. K., Coleman, C., Garcia, H. E., Grodsky, A., Johnson, D. R., Locarnini, R. A., Mishonov, A. V., O'Brien, T. D. and others: World Ocean Database 2013., 2013.

Breitburg, D., Levin, L. A., Oschlies, A., Grégoire, M., Chavez, F. P., Conley, D. J., Garçon, V., Gilbert, D., Gutiérrez, D., Isensee, K., Jacinto, G. S., Limburg, K. E., Montes, I., Naqvi, S. W. A., Pitcher, G. C., Rabalais, N. N., Roman, M. R., Rose, K. A., Seibel, B. A., Telszewski, M., Yasuhara, M. and Zhang, J.: Declining Oxygen in the Global Ocean and Coastal Waters, Science, 359(6371), eaam7240, doi:10.1126/science.aam7240, 2018.

Breitburg, D. L., Loher, T., Pacey, C. A. and Gerstein, A.: Varying Effects of Low Dissolved Oxygen on Trophic Interactions in an Estuarine Food Web, Ecological Monographs, 67(4), 489–507, doi:10.2307/2963467, 1997.

Capet, X., McWilliams, J., Molemaker, M. and Shchepetkin, A.: Mesoscale to Submesoscale Transition in the California Current System. Part I: Flow Structure, Eddy Flux, and Observational Tests, Journal of Physical Oceanography, 38(1), 29–43, 2008.

Capet, A., Meysman, F.J.R., Akoumianaki, I., Soetaert, K., Grégoire, M.: Integrating sediment biogeochemistry into 3D oceanic models: A study of benthic-pelagic coupling in the Black Sea. Ocean Modelling 101, 83–100. doi.org/10.1016/j.ocemod.2016.03.006, 2016.

Castaing, B., Gagne, Y. and Hopfinger, E. J.: Velocity Probability Density Functions of High Reynolds Number Turbulence, Physica D: Nonlinear Phenomena, 46(2), 177–200, doi:10.1016/0167-2789(90)90035-N, 1990.

Castro, C.G., Nieto-Cid, M., Álvarez-Salgado, X.A., Pérez, F.F.: Local remineralization patterns in the mesopelagic zone of the Eastern North Atlantic, off the NW Iberian Peninsula. Deep Sea Research Part I: Oceanographic Research Papers 53, 1925–1940. doi.org/10.1016/j.dsr.2006.09.002, 2006.

[revised manuscript text omitted]

*influences the oxygen cycle, related to processes in the benthic layers, and on remineralization processes that influence the oxygen cycle, and that are not referred to.*

Authors: *We are grateful to the referee for bringing to our attention to this interesting body of previous research. We indeed referred to Rossi et al.'s articles since the dataset of this specific campaign was used to tune our model (e.g. nitrate/temperature relationship) and to compare its simulated outputs to in-situ observations. Note also that some of the authors suggested by referee 2 were already cited in our original manuscript (e.g. Alvarez-Salgado et al. 1993; Reboreda et al. 2014; 2015). Following his/her suggestions, we have now included additional references to Pérez et al (2001), Castro et al (2006) and Alvarez-Salgado et al. (2013) as follows:*

"During the upwelling season, hydrographic and biogeochemical variability is primarily determined by the wind forcing that controls the inflow of offshore deep water masses onto the shelf (Alvarez-Salgado et al., 1993). DO levels in these subsurface waters are thought to diminish during the upwelling season due to strong remineralization of sinking organic matter as well as to mixing with poorly-ventilated waters of subtropical origins (Castro et al., 2006). Overall, DO appears to partly control the remineralization of dissolved organic carbon in the IPUS and more generally in key water masses of the North Atlantic (Alvarez-Salgado et al. 2013)."

"These structures have a strong influence on the cross-shore transport and on the vertical redistribution of biogeochemical tracers (Bettencourt et al., 2017; Combes et al., 2013; Gruber et al., 2011; Hernández-Carrasco et al., 2014; Nagai et al., 2015; Renault et al., 2016; Rossi et al., 2013). In the Iberian basin, the Eastern North Atlantic Central Water (ENACW) mass of subtropical origin, whose archetypal concentrations of DO are initially low, would be ventilated by eddy-induced mixing with more oxygenated ENACW of subpolar origin (Pèrez et al., 2001)."

*Concerning the second part of this comment, we agree with referee 2 that our O2PZ model disregards, as a result of its low complexity, some oceanographic processes that influence oxygen cycling over the shelf, such as complex processes occurring within the benthic boundary layers or into the upper sedimentary layers (i.e. the so-called benthic-pelagic coupling). We now acknowledged this limitation of our idealized modeling approach in the revised Discussion and Conclusions section (Section 4, 1$^{st}$ paragraph):*

"We built a coupled physical-biogeochemical model of an idealized seasonal coastal upwelling to study the effect of the upwelling season length and phytoplankton community structure on dissolved oxygen inventory. The low-complexity $O_2PZ$ model accounts for oxygen production by photosynthesis and consumption by both respiration and remineralization but its simplicity does not allow to include other oceanographic processes that could influence oxygen cycling over shallow shelves, such as for instance the complex benthic-pelagic coupled processes (Capet et al., 2016)."

Ref. #2: *3 - Lines 62-64 need a reference that support those statement.*
Authors: We have added appropriate references to support those statements.

"Moreover, diatoms growth is thought to be positively correlated with the upwelling-driven nitrate inputs (Sarthou et al., 2005); in contrast, the growth of other planktonic groups can be insensitive to and even limited by newly-upwelled nitrates (Mahaffey, 2005), especially when colimitation prevails or in the absence of necessary micro-nutrients."

Ref. #2: *4 - Is the implementation of the Biogeochemical (BGQ) module done in CROCO model, or was implemented from scratch?. This is not clear in the ms. In that case, justify why none of the other BGQ modules was not used.*

Authors: *Our low-complexity model centered on oxygen was implemented from scratch in the CROCO model, i.e. it was hard-coded into the CROCO model following, as a template, one of the already-embedded BGQ modules. The O2PZ model was retained instead of other BGQ models available in CROCO because of its low complexity (only 3 equations) resulting in fast and efficient computing, hence allowing running/analyzing multiple simulations while considering the role of high-resolution dynamics (e.g. submesoscale) for oxygen cycling. It has been specified in the manuscript (beginning of sect. 2.1):*

"The low-complexity $O_2PZ$ model was directly implemented into the CROCO model following one of the already-embedded biogeochemical modules; it allows maintaining the full capabilities and versatility of this community code while gaining in computing efficiency (allowing extensive sensitivity studies and the consideration of submesoscale dynamics in oxygen cycling) as compared to the more complex biogeochemical modules already available within CROCO."

Ref. #2: *4- Is not clear the interaction with the atmosphere in the BGQ module, concerning O2 atmosphere - ocean exchange.*

Authors: *The air-sea fluxes of DO follow the formulation of Wanninkhof (1992). The flux is Va\*([O2]-[O2]$_{sat}$), where Va is the gas piston velocity, [O2] is the DO concentration at the water surface (the first sigma level below the surface) and [O2]$_{sat}$ is the O2 solubility in seawater. Please, see Equation 1 and the description of the DO gas exchange in the revised version of the manuscript (end of $2^{nd}$ paragraph in page 4):*

"On top of the oxygen that is produced and consumed within marine ecosystems, a considerable part of dissolved oxygen eventually becomes gaseous and is then exchanged at the ocean-atmosphere interface, contributing to the estimations that more than one half of atmospheric oxygen is produced in the ocean (Harris, 1986).The remaining term of Eq. (2) is the air-sea flux of $O_2$, which is the product of the gas transfer velocity velocity *Va*, that depends on the square of the wind speed and on the Schmidt number (Wanninkhof, 1992), with the difference of the oceanic DO concentration and the solubility of $O_2$ in seawater."

Ref. #2: *5- Laplacian mixing along sigma surfaces tends to generate spurious currents. Why you did not use (ro- tated) mixing along Geopotential?*

Authors: *The reason for not using mixing along Geopotentials was that this option is computationally heavier than Laplacian mixing along sigma surfaces, while their methodological biases are similar. We retained the option that resulted in a faster code without compromising the quality of our analyses.*

Ref. #2: *6 - Is the wind only alongshore?, please specify.(l 109).*

Authors: *Yes. In our idealized setting the wind is only alongshore. Given the idealization level, we didn't see the need to include an across-shore wind component. We modified the text above Table 1 to make this explicit:*

"The cross-shore wind component is zero in all cases."

Ref. #2: *7 - It should be clearer to the reader why nutrients are not introduced specifically, and what is the advantage of doing it in an O2PZ model instead of O2NPZ?*

Authors: *The addition of a nutrient compartment to the O2PZ model would make it computationally heavier and we wanted to keep the calculations as simple and as fast as possible. Thus, given that the original formulation of the O2PZ model didn't have nutrient limitation mechanism, which is important for upwelling regions such as the IPUS. As density and nitrate are closely related, especially in upwelling regions, we came up with a relatively simple way to include this nutrient limitation effect in our oxygen centered model. Please, see the revised text in the last paragraph of page 4:*

"This term [l(N)] is included here to circumvent the absence of nutrient limitation in the original $O_2PZ$ model of SP2015. We chose to add a parametrization instead of a new equation for the nutrients because this last option would necessarily increase the complexity of the model."

Ref. #2: *8 - The 12m/s wind pulses although exist in particular years (like the MOUTON campaign) are too high and unrealistic for the Western Iberia region. Justify better why such high values. Some sensitivity studies was done to this value?*

Authors: *Our choice of the wind pulse was based on forcing the model with a wind whose magnitude is comparable (i.e. the same order of magnitude, bearing in mind that our configuration is highly idealized) to the one observed during the MOUTON campaign. We also retained the highest magnitude (rather than the mean or lowest) because it stimulates the quickest and clearest formation of the unstable upwelling front. Note however that preliminary analyses were carried out to test several wind speeds (not shown); we found that the model responses were qualitatively similar, provided that the magnitude used is above a certain threshold, ensuring the development of upwelling (e.g. isopycnal outcropping and subsequent front destabilization). Given those results, we decided to use the wind speed that represented best the MOUTON campaign forcing conditions. We have clarified our rationale behind these choices in sect. 2.2 of the revised manuscript:*

"The cross-shore wind component is zero in all cases. Despite the fact that "true" wind patterns observed over Western Iberia are undoubtedly more variable (both in terms of direction and magnitude) than our idealized forcing inspired from a specific campaign, it provides the quickest development of a clear upwelling front generating small-scale instabilities. Note that sensitivity tests were performed (not shown) and revealed that the model responses are qualitatively similar among different wind strengths, given that the magnitude is high enough to ensure the development of the unstable upwelling front."

Ref. #2: *9- I don't understand why in the initial vertical profile, (Fig 3a) the 02 drops to zero when in reality the O2 at 300m is closer to 230 mmol O / m3. The range of values in the region above central waters is 220-290 mmol O / m3.*

Authors: *The drop in O2 levels is a consequence of our very idealized setting. Indeed, as the referee rightly points out, O2 levels in the region at 300 meters are higher. However, in our highly idealized setting, we do not have subsurface O2 inputs due to the periodic boundary conditions at the North and South boundaries. Since in the O2PZ model, the only O2 source is photosynthesis,*

*that is zero below the euphotic zone, without the subsurface O2 inputs, the subsurface O2 becomes depleted by respiration and remineralization. Note that this point (absence of subsurface O2 input) was already reported as an important point of discussion that we clearly identified as key for future work (please see sect. 4).*

Ref. #2: *10 - Claim that figure 4 represents a validation seems too optimistic to me. I can't understand what the red dot represents in the figures.*

Authors: *Please, see the reply to point 11 below. The red dots are used to mark the position of the barycenter, i.e. the average values of the data for each quantity represented in the scatter plots, i.e. the x-coordinate of the red dot in figure 4a is the averaged density and the y-coordinate is the mean [O2].*

Ref. #2: *11 - Using logarithms in Chla completely distorts the comparison, because the observations are showing values of 0-10 mg Chla.m-3 between isopycn 26-27 as the model rarely goes beyond 1 (at isopycn 27). Similar comment arises for O2. If you claim those results are validated, then, any result is validated. If you insist in keeping this part as it is, I suggest that you remove the word validation, perhaps use 'qualitative comparison'.*

Authors: *We agree that the use of logarithmic scales provides mainly a qualitative comparison between model results and observations; note however that Chlo-a concentrations are commonly plotted using logarithmic scales as values often span several order of magnitude, as is the case here. Nevertheless, we have modified the text to make this clear, removing all references to validation. Please, see the revised text of the last paragraph of section 2.*

"We compared the full ranges of densities $\rho$, DO concentrations [$O_2$], and chlorophyll-a concentrations [Chl-a] of the reference simulation to all compiled measurements collected during the MOUTON campaign. Given the low-complexity of the biogechemical model and the highly idealized physical setting, a fully quantitative comparison is out of scope, however we confirm that the coupled model reproduces qualitatively well the $\rho$-[$O_2$] and $\rho$-[Chl-a] relationships obtained from in-situ observations."

Ref. #2: *12-Figure 5, I can understand the effect of upwelling at x= 40km that is related to shelf upwelling, but how do you interpret the dome at 60-70 km (approximately)? To use the references (l 200-204) to validate the averages of O2 seems to me to be an over-optimistic approach. Again you can say that it is a simplified and idealized model, but do not try to convince the reader that it is validated, because readers with observational background would hardly understand it.*

Authors: *We believe the dome at 60-70 km is caused by the downwelling signal induced by the eddy induced circulation. More specifically, this downwelling divides the region of recently upwelled low DO waters into (i) the inshore low DO region (within 50 km from the shores) and (ii) the dome located at 60 – 70 km, slightly off the shelf edge. We have modified the text to highlight that our results are similar in a qualitative sense. The revised text ($1^{st}$ paragraph of section 3.1) now reads:*

"In a qualitative sense our idealized model results are similar to more realistic model studies such as , Gutknecht *et al.* (2013) which in their modelling study of the Benguela upwelling region found a low [$O_2$] plume for the climatological month of December at the shelf edge, and to observational studies such as Hales *et al.* (2006), which for the Oregon coast, measured [$O_2$]

of 70-110 mmol m$^{-3}$ in upwelled water at the shelf break, about 200 mmol m$^{-3}$ less than at the surface, which is the same range of the vertical gradient simulated here."

Ref. #2: *13- Lines 210 - 215 seem more like a discussion, and their presence is not understood there.*

Authors: *Lines 210-215 provide a necessary, yet short, physical description of the simulation dynamics. It is needed to put the average DO distribution in the context of upwelling dynamics (at very first order, any oceanic tracer is influenced by the ocean circulation). Particularly, we felt that a discussion of the mean and the eddy-induced circulation was required to better explain the patterns of nearshore DO seen in Figure 5 (see also the first part of our reply to point 12: the "dome" is well explained by the eddy induced circulation which has been described just before).*